# Upland Yedoma taliks are an unpredicted source of atmospheric methane

K. M. Walter Anthony [1] ✉, P. Anthony[1], N. Hasson[1], C. Edgar [2], O. Sivan[3],
E. Eliani-Russak[3], O. Bergman[1,3], B. J. Minsley [4], S. R. James [4], N. J. Pastick[5],
A. Kholodov [6], S. Zimov [7], E. Euskirchen [2], M. S. Bret-Harte [2],
G. Grosse [8,9], M. Langer[8,10] & J. Nitzbon [8]

Landscape drying associated with permafrost thaw is expected to enhance microbial methane oxidation in arctic soils. Here we show that ice-rich, Yedoma permafrost deposits, comprising a disproportionately large fraction of pan-arctic soil carbon, present an alternate trajectory. Field and laboratory observations indicate that talik (perennially thawed soils in permafrost) development in unsaturated Yedoma uplands leads to unexpectedly large methane emissions (35–78 mg m$^{-2}$ d$^{-1}$ summer, 150–180 mg m$^{-2}$ d$^{-1}$ winter). Upland Yedoma talik emissions were nearly three times higher annually than northern-wetland emissions on an areal basis. Approximately 70% emissions occurred in winter, when surface-soil freezing abated methanotrophy, enhancing methane escape from the talik. Remote sensing and numerical modeling indicate the potential for widespread upland talik formation across the pan-arctic Yedoma domain during the 21$^{st}$ and 22$^{nd}$ centuries. Contrary to current climate model predictions, these findings imply a positive and much larger permafrost-methane-climate feedback for upland Yedoma.

Increases in arctic air temperature nearly four times faster than the global average[1] are leading to widespread warming and thaw of permafrost soils[2,3] often accompanied by drying of terrestrial ground surfaces[4,5]. A long-term trajectory of thaw-driven landscape drainage and drying across much of the Arctic is expected, despite predictions for increased precipitation[6–8]. Soil warming is greatest in places where ground temperatures have historically been lowest - in continuous permafrost regions such as Northern Siberia. However, in discontinuous permafrost areas, recent warm winters and high snowfalls have already caused widespread talik development when seasonal surface freeze no longer extends down to the top of the permafrost table[9].

Permafrost thaw and talik formation are a climate concern[10] given the potential release to the atmosphere of the immense permafrost soil organic carbon pool [~1600 petagrams, (Pg) C], much of which has been out of circulation with the atmosphere (~750 PgC) for thousands of years[11,12]. Upon thaw, this soil organic carbon (C$_{org}$) is mineralized by microbes that generate carbon dioxide (CO$_2$) and methane (CH$_4$)[13]. When these old ($^{14}$C-depleted) decomposition products enter the atmosphere, they perturb the existing carbon balance, offsetting vegetative uptake of carbon dioxide[10,14]. Formation of $^{14}$C-depleted, methane-emitting thermokarst (thaw) lakes[15,16] and other abrupt-thaw, wetland-type features[17,18] are considered positive feedback to climate warming alongside other natural and anthropogenic emission sources[19].

[1]Water and Environmental Research Center, University Alaska Fairbanks, Fairbanks, AK, USA. [2]Institute of Arctic Biology, University Alaska Fairbanks, Fairbanks, AK, USA. [3]Department of Earth and Environmental Sciences, Ben Gurion University of the Negev, Beersheva, Israel. [4]U.S. Geological Survey, Geology, Geophysics, and Geochemistry Science Center, Denver, CO, USA. [5]U.S. Geological Survey, Earth Resources Observation and Science Center, Sioux Falls, SD, USA. [6]Geophysical Research Institute, University Alaska Fairbanks, Fairbanks, AK, USA. [7]Pacific Geographical Institute of the Russian Academy of Sciences, Northeast Science Station, Cherskiy, Russia. [8]Alfred Wegener Institute Helmholtz Centre for Polar and Marine Research, Potsdam, Germany. [9]University of Potsdam, Institute of Geosciences, Potsdam, Germany. [10]Department of Earth Sciences, Vrije Universiteit Amsterdam, Amsterdam, The Netherlands. ✉e-mail: kmwalteranthony@alaska.edu

Recent research focused on uplands has dampened the alarm for permafrost-methane emission this century. Loss of lakes[4] and ice-wedge degradation[5] lead to the drying of arctic soils, a process that reduces water-filled pore space and enhances oxygen and methane diffusion in active layer soils. Consequently, upland soils are expected to increase their sink capacity for methane, both by uptake of atmospheric methane and as a biofilter when microbially produced methane migrates from water-saturated, anoxic soil layers that occur above the permafrost table in some areas[20–22]. Oh et al.[23] predicted that high-affinity methanotrophs (HAMs), which are methane-oxidizing bacteria living in upland arctic soils, will compensate for most of the projected rise in 21st-century wetland emissions associated with arctic permafrost thaw. Accordingly, upland environments with well-aerated surface soils, environments that are today a minuscule source of atmospheric methane in tundra ($-1$ to $2\,\mathrm{mg\,CH_4\,m^{-2}\,d^{-1}}$) and a sink in boreal forests ($-1$ to $-0.2\,\mathrm{mg\,CH_4\,m^{-2}\,d^{-1}}$)[24], are expected to become an increasingly important methane sink and negative feedback to climate warming in the future[22,23].

Dry upland soils spatially dominate the 17.8 million km² permafrost region[11,18,25], but so far, permafrost carbon models of methane dynamics in upland environments do not distinguish among permafrost soil types or account for winter emissions[10,14,18,26]. About three percent of the permafrost region (0.48 million km²) is classified as intact, undisturbed Yedoma, defined as thick ($\leq 50\,\mathrm{m}$) silt-dominated,

ice-supersaturated permafrost soils occurring in uplands and hill areas that have remained frozen since their initial formation in unglaciated mammoth-steppe tundra regions of Siberia, Alaska and Northwest Canada during the late Pleistocene[27,28]. Thinner and/or more segmented Yedoma deposits occur in various degraded and undegraded forms across the entire Yedoma domain (2.5 million km², Fig. 1). Despite its small area, the Yedoma domain contains a disproportionately large quantity of soil organic carbon, 327–466 Pg[27], which is >25% of the total northern permafrost soil carbon pool (1307 Pg)[11]. Unlike most permafrost uplands where soil organic carbon is limited to the surface three meters[11], Yedoma deposits extend tens of meters belowground. The unique thickness and paleoenvironmental factors regulating soil carbon availability[27] imply that methane cycling in Yedoma taliks may deviate from the non-Yedoma upland permafrost environments studied so far[20,24,29–32].

Land surface models predict minor methane emissions from all 21st-century uplands since aerobic rather than anaerobic processes are expected to dominate carbon cycling in well-drained environments[14,18,26]. However, if Yedoma uplands were found to become anaerobic and strong sources of atmospheric methane upon thaw[14,18,26], rather than insignificant sources or sinks[24], the permafrost carbon feedback could be substantially larger given the 25–34 times higher global warming potential of methane relative to carbon dioxide on the century time scale[33].

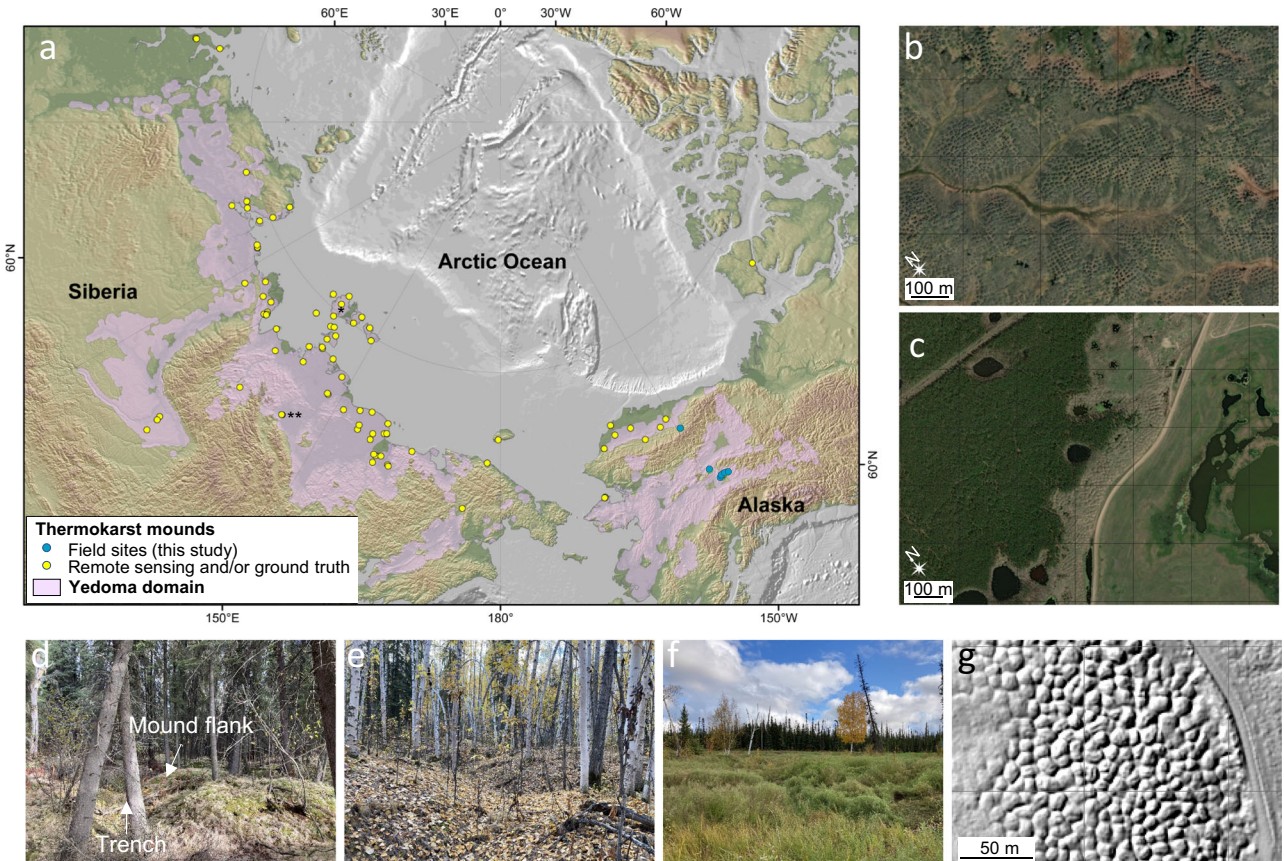

**Fig. 1 | Upland thermokarst mounds in the pan-arctic Yedoma domain. a** Map showing select locations where thermokarst mounds (thaw features) have been observed in remote sensing optical satellite imagery and/or ground truth (yellow dots; Table S2) and in our Alaska fieldwork (blue dots; Fig. S1). Examples of thermokarst mounds in satellite imagery are shown for (**b**) Kotelny Island, Northeast Siberia (* in **a**) and **c** Tabaga, Central Yakutia (** in **a**). Panels **d–f** are photos of thermokarst mounds in interior Alaska spruce forest (**d**), deciduous forest (**e**), and grassland (**f**) ecosystems. **g** High-resolution thermokarst-mound topography below forest canopy as reconstructed from airborne LiDAR data from central Alaska[41]. Mound spacing is typically ≤15 m, following Pleistocene ice-wedge polygon patterns[37]; vertical thermokarst-mound relief can vary from less than one meter to five meters. Yedoma extent (pink) in **a** is from Strauss et al.[28] and background topography and bathymetry data is based on NOAA National Geophysical Data Center[91]. In (**g**), the Alaska Division of Geological and Geophysical Surveys granted permission to use the elevation data, which is in the public domain.

Using a combination of field-flux measurements at 26 boreal and arctic sites in Alaska, radiocarbon dating, geophysics, and soil geochemical and microbial profiles ("Methods" section), we show that upland forested and grassland hillslopes with thermokarst features initiated in the last 40–70 years by the thawing of ice-rich Yedoma permafrost are surprisingly an exceptionally large source of in situ microbially produced methane. Methanogenesis in thawing Yedoma permafrost is active year-round. Further, we show that methane emission is elevated in winter when surface-soil freezing impedes methanotrophy[34] and extreme wintertime barometric pressure fluctuations pump methane[35] out of taliks through frost-induced cracks. Finally, remote sensing and numerical modeling indicate that across the pan-Arctic, Yedoma uplands are susceptible to talik formation in the 21st and 22nd centuries, even in Northern Siberia where soils are currently coldest[2]. These findings have important implications for the permafrost carbon feedback. Talik development in well-drained Yedoma uplands should be considered as a unique landscape process with potentially very large permafrost-soil-derived methane emissions, particularly in winter.

## Results and discussion
### Methane emissions from thawing Yedoma uplands

We conducted >1200 extensive, year-round chamber-flux measurements from spring 2020 to summer 2023 at 26 forested, grassland, and tundra upland sites in Alaska characterized by thermokarst mounds (Figs. 1 and S1) and at 16 upland control sites lacking thermokarst mounds (Table S1) ("Methods" section). Thermokarst mounds, termed baidjarakhs in Yakutia[36], are regularly spaced conical hills (≤15 m diameter, ≤5 m height) separated by trenches (≤3 m width) that form in degrading ice-rich Yedoma permafrost environments[37,38]. Their formation and morphology are based on the melting of large syngenetic

ice wedges in polygonal patterned ground, where the polygon margins (trenches) underlain by ice wedges subside faster and deeper than the less ice-rich polygon centers (mound tops), leaving behind distinct conical-mound features in regularly spaced patterns (Fig. 1). Recently recognized as sources of nitrous oxide[39], thermokarst mounds have been detected in high-resolution optical satellite imagery, airborne LiDAR imagery, and/or fieldwork across the entire Yedoma domain (Fig. 1, Table S2). These mounds form in locations where surface soil overlying intact Yedoma has been disturbed by wildfire, slope or fluvial erosion, thermokarst, and land clearing[36,37,40]. Our interior Alaska field sites were identified a priori in airborne laser scanning LiDAR elevation data[41], in which mounds formed by degrading Yedoma permafrost are visible (Fig. 1g). In the Alaskan tundra, where we lacked LiDAR data, thermokarst-mound topography was identified during fieldwork.

In addition to chamber-flux measurements at extensive sites, we conducted intensive measurements using eddy covariance, radiocarbon dating, geophysics, and borehole drilling at North Star Yedoma (NSY, informal name), a 5-ha north-facing gradually-sloped field (21:1 slope ratio) underlain by thawing, ice-rich Yedoma permafrost in interior Alaska (Fig. 2). At NSY, native grasses interspersed with moss (Fig. 2d) established on the surface of thermokarst mounds that developed over decades following 1970s clearing of the mature black spruce forest, insulating surface moss, and surface organic soils. NSY was operated as a rugged golf course for nearly two decades until 2021 when the field was left alone; mowing resumed in summer 2022 under new property ownership. Geophysical surveys at NSY revealed a spatially varying ~5 to 9 m thick talik (Fig. S2), confirmed by borehole drilling (6.8–7 m to permafrost) and year-round soil temperature monitoring. Direct observations showed talik soils were mostly water-unsaturated, including at the base of the talik (Fig. 2f). A 3-ha triangular plot adjacent to our grass-dominated study site underwent the same

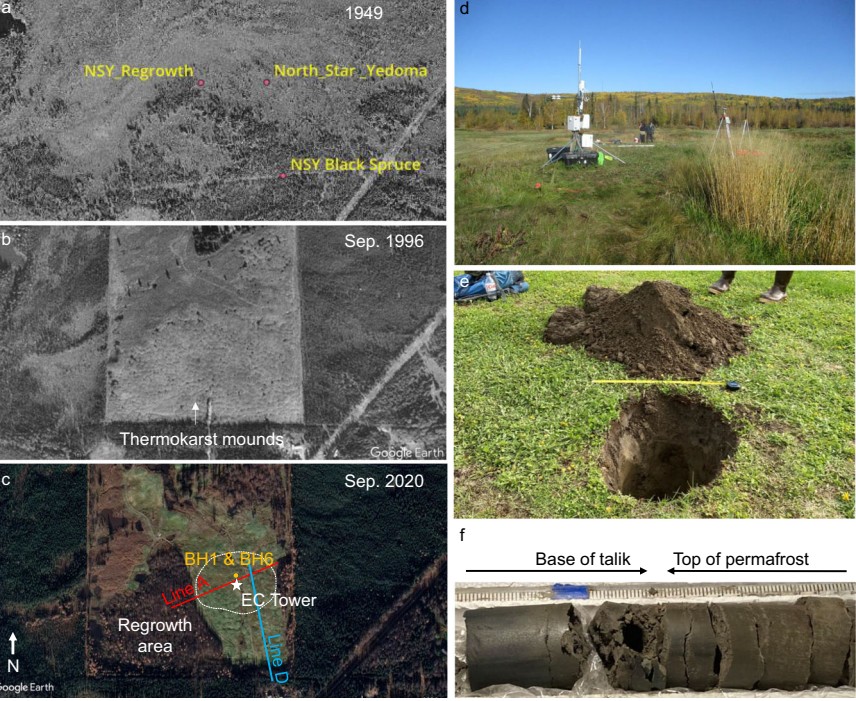

**Fig. 2 | Chronological images and field photos of the North Star Yedoma (NSY) study site. a** 1949 single-band airborne optical image available through USGS Earth Explorer (https://pubs.usgs.gov/gip/AerialPhotos_SatImages/aerial.html). **b** 1996 image showing widespread thermokarst-mound formation. **c** 2020 image showing deciduous forest succession in the western regrowth section and grass in the eastern section. The timing of the initial disturbance (sometime between August 1976 and August 1978) was determined by comparing Landsat false color

composites that indicated disturbance to vegetation within this time frame. Locations of the eddy-covariance tower (EC; 80% footprint white dotted line), geophysical observations (Lines A and D), and boreholes (BH) are shown in (**c**). **d**–**f** Ground photos of the EC tower in the study field (**d**), soil pit (**e**), and lowest portion of the 7-m-long soil core (**f**). Panels (**e**) and (**f**) show dry soil conditions near the ground surface [22–26% volumetric water content (VWC)], the base of the talik (26–30% VWC), and top of permafrost (21–26% VWC).

initial disturbance, but a lack of human activity during the past 45 years led to natural succession forest regrowth atop well-developed thermokarst mounds (Fig. 2c).

The accumulation of flammable gas pockets trapped in the sod layer at NSY was first noticed in 2018[42]. The phenomenon was similar to observations in the Siberian tundra in 2016[43]. When pressed from above, methane-containing turf bubbles (30–60 cm long axis) moved laterally in the void between the organic soil and sod layer, resembling waterbed fluidity, except in this case, the fluid was gas, not water. Turf bubbles tended to occur after spring and autumn rain events, and were focused along the toe slopes of thermokarst mounds where the interface between degrading ice wedges in trenches and relatively ice-poor mound soils creates conduits (e.g., microcracks, fissures, and desaturated pore-space channels) for gas migration and escape. Turf-bubble gas sampled from two NSY thermokarst mounds in September 2018 ("Methods" section) was dominated by methane ($CH_4$, $62 \pm 16\%$ by volume; $CO_2$, $2.8 \pm 1.4\%$) with $\delta^{13}C_{CH4}$ values ($-67.0 \pm 3.9‰$) consistent with a microbial origin. The radiocarbon age of turf bubble methane ($^{14}C_{CH4}$ $6650 \pm 15$ yr BP) was similar to that of plant macro-fossils ($^{14}C_{org}$ 7770–8210 yr BP) extracted near the base of a 3.2-m soil core (BH1 in Fig. 2c), but nearly twice as old as the radiocarbon-dated carbon dioxide in the same sample ($^{14}C_{CO2}$ $2885 \pm 15$ yr BP). This indicates either an older permafrost soil organic carbon substrate fueling methanogenesis compared to carbon dioxide production, or dilution of old carbon dioxide in bubbles by younger sources.

Compelled by the discovery of methane in a well-drained, grassland field, we conducted chamber-based surveys of methane fluxes at 26 thermokarst-mound sites across Alaska that varied according to latitude, time since disturbance, and present-day vegetation cover to determine how widespread methane emissions are among thermokarst-mound uplands (Fig. S1, Table S1a). In 23 out of the 26 sites, we observed methane emissions of various magnitudes (Fig. 3, Table S1b), while all 16 control sites (lacking thermokarst mounds)

exhibited negative or net zero methane fluxes. The three thermokarst-mound sites with either net negative or neutral fluxes were located on the University of Alaska Fairbanks (UAF) campus in areas that were disturbed as far back as 1908 (Table S1a), and which today are dominated by mature deciduous or mixed forests. It is possible that permafrost carbon mineralization in taliks diminishes in upland sites on the century time scale, a phenomenon observed in thermokarst lakes[44], such that older taliks produce less methane; however, other factors such as evapotranspiration and landscape position also likely affect the balance of methane production and consumption in upland soils.

Methane emissions were higher from thermokarst mound sites compared to controls (Table S1b). Fluxes at NSY fell within the range of 23 other methane-emitting Alaskan upland thermokarst-mound sites (Fig. 3a, b, e; Table S1b) [NSY: $110 \pm 24$ mg $CH_4$ $m^{-2}$ $d^{-1}$, mean ± standard error of the mean (SEM), $n = 665$; Extensive sites: $87 \pm 38$ mg $CH_4$ $m^{-2}$ $d^{-1}$, $n = 450$]. To further explore the potential for surface-soil disturbance and vegetation management to impact methane fluxes, we compared a different thermokarst-mound grassland site heavily disturbed in 2023 to the adjacent, undisturbed thermokarst mounds and found no significant differences (Supplementary Results 4.1). Our findings indicate that methane emissions are widespread among upland taliks formed within the last 40–70 years and that NSY is not an outlier study site.

## Hydrology and hotspots

At our 25 extensive thermokarst-mound study sites and NSY, methane flux was weakly related to surface soil moisture (thermokarst mounds: Pearson correlation coefficient $r(273) = 0.33$, $p < 0.001$; NSY: $r(314) = 0.51$, $p < 0.001$; Supplementary Results 4.2) and field-scale microtopography (Table S3). The extensive thermokarst-mound study sites had similar methane fluxes as NSY, but over a much broader range of surface soil moisture conditions, with substantially positive

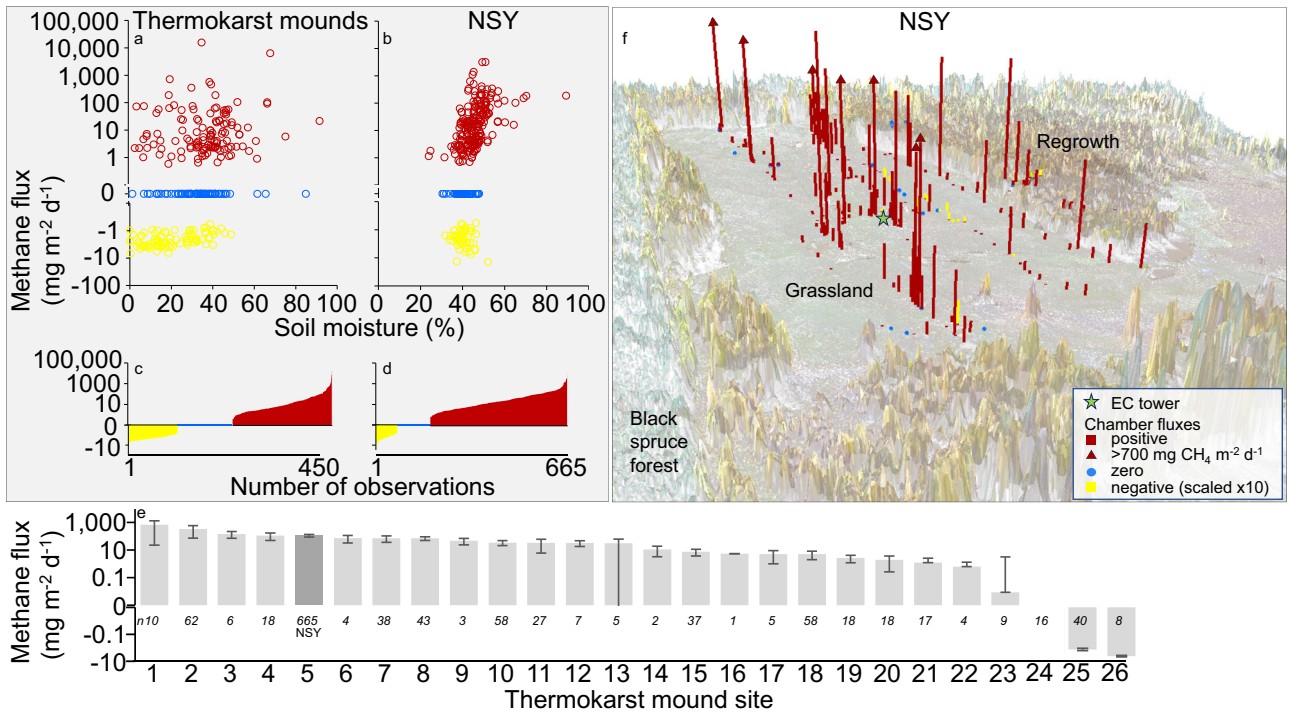

**Fig. 3 | Plot-scale methane fluxes and soil moisture at 25 upland thermokarst-mound study sites with boreal forest, grassland, and tundra vegetation and NSY, our intensive thermokarst-mound study site. a, b** Methane fluxes in relation to surface (12-cm) soil moisture. **c, d** Summary of all chamber-flux measurements in thermokarst-mound uplands. **e** Thermokarst-mound methane fluxes at specific sites for biological replicates, the sample size of which is indicated by *n* (gray bars: mean, SEM) (Table S1). **f** Plot-scale mean methane emissions <700 mg $m^{-2}$ $d^{-1}$ at NSY [Emissions ≥700 mg $m^{-2}$ $d^{-1}$ are indicated with arrows and negative (uptake) emissions are multiplied by 10 for visualization].

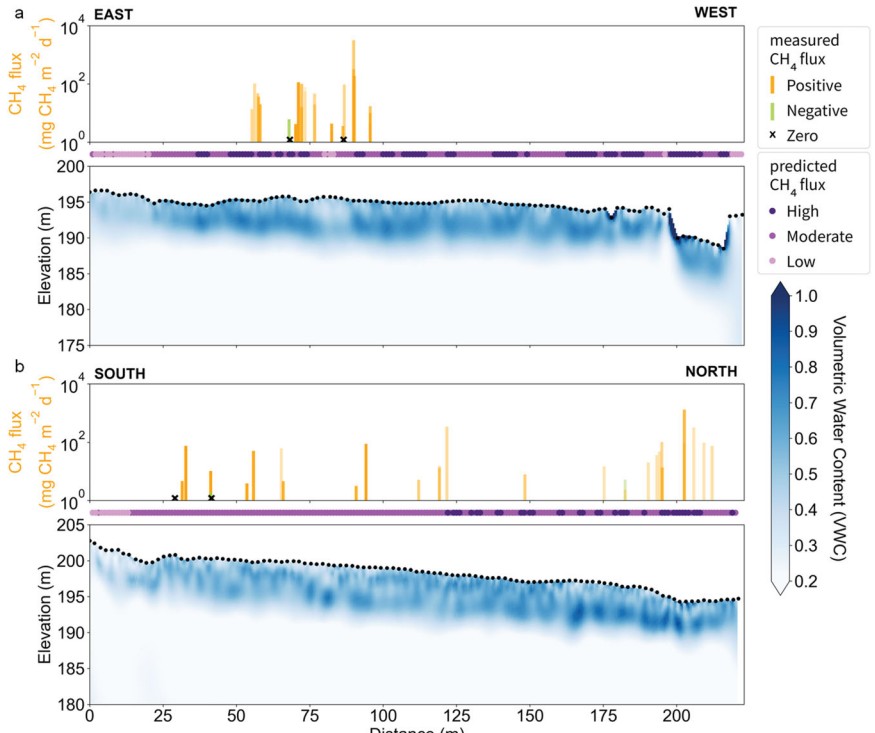

**Fig. 4 | Geophysically derived water content with observed fluxes and predicted flux classes at NSY in September 2021.** Electrical Resistivity Tomography (ERT) transects transformed to VWC estimates for the NSY E−W transect A (**a**) and N−S transect D (**b**) profiles are shown together with observed methane (CH₄) fluxes within 5 m (transparency scaled by distance from ERT line) and predicted flux classifications (purple ribbon plot) at continuous locations along each profile.

methane fluxes (>50 mg $CH_4$ $m^{-2}$ $d^{-1}$) observed at VWC as low as <5%. This atypical pattern of high emissions from dry surface soils reveals that the soil moisture regime and gas transport pathways in meters-thick, thawed Yedoma silt are more nuanced than in arctic wetlands and other upland-soil active layers, environments where the water table (and soil carbon) is closer to the ground surface, controlling methane fluxes through the regulation of oxygen and methane diffusion rates[6,32,45–48].

To gain a better understanding of the two-dimensional soil moisture regime at the field scale, we conducted geophysical surveys of electrical resistivity along ~200-m-long transects coupled to down-hole nuclear magnetic resonance measurements (Supplementary Methods 3.1). NSY soils were characteristically dry near the surface [volumetric water content (VWC) 43.9 ± 0.4%, mean ± SEM, $n = 316$, Fig. 3b], but variably wetter at intermediate depths, particularly lower on the gradually-sloping field (Fig. 4). We assessed the relationship between plot-scale methane fluxes and microtopography with estimates of resistivity and soil VWC using a regression model (Supplementary Discussion 5.3). Due to the low number of zero and negative fluxes recorded within 2 m of the ERT transects−$n = 4$ and $n = 2$, respectively−relationships were dominantly driven by the positive methane fluxes. This analysis confirmed a complex relationship between vertical soil moisture profiles and chamber-measured fluxes (Fig. S3). Drier mound tops corresponded to low flux, and wetter areas were generally associated with higher fluxes (see also Table S3), with the maximum water content in the upper 3 m being the primary driver of the positive correlation to methane fluxes observed. However, by accounting for the relative vertical water content profile at any location in addition to the maximum water content, we determined that wetter conditions at intermediate depths in the soil profile beneath drier surface soils potentially lead to greater methane emission (Fig. 4). This relationship suggests atmospheric fluxes may be enhanced when conditions support both preferential flow pathways of gas escape through drier shallow soils along with wetter, deeper methane-producing soils that can outpace methanotrophy. That is, shallow (<1 m) soil moisture patterns alone may not fully capture the range of conditions important for the generation and surface flux of methane, especially in deeply thawed Yedoma environments. Furthermore, the geophysical results indicated that water saturation varies spatially with an inverse correlation to topography, i.e. higher water content in topographic lows. These water content oscillations have a spatial periodicity (distance over one peak-trough cycle) that maintains a long-period signal throughout the talik zone on a scale matching that of the thermokarst mounds (Fig. S4). Shorter distance oscillations are also seen across shallow depths (0–2 m) but disappear or lengthen at greater depths. Together these patterns in subsurface water content suggest infiltration flow paths and saturation at depth may be guided preferentially by the imprint of current or recent ice wedges.

The hypothesis that preferential methane flow pathways exist in thawed Yedoma silts was supported by our observation of hotspot emissions at NSY at all times of year and at all field elevations (Fig. 3f). Among 508 chamber-flux measurements, in which microtopographic position was well-constrained, 74 outlier hotspot fluxes (89–1386 mg $CH_4$ $m^{-2}$ $d^{-1}$) comprised 15% of the measurements but 92% of the total flux. Half of these hotspots ($n = 38$) occurred on dry mound flanks.

The prevalence of methane seeps on mound flanks was also observed in thermokarst lakes[49] and can be attributed to Yedoma permafrost soil history. At the time of original syngenetic permafrost formation during the last glacial period, Yedoma ecosystems were dominated by productive grasses growing in environments of rapid silt deposition. Seasonal freeze-thaw cycles and segregated ice-lens formation created platy soil structure in the otherwise massive silt, rich in grass roots[38]. Today upon thaw, decomposition of these ancient grass roots creates voids in which excess gas can accumulate. Melted ground-ice layers (post-cryogenic structures) become routes for gas migration and subsequently are most likely to adjoin mechanical

cracks along the interfaces of ice-wedge casts (imprints of former ice wedges) forming preferential gas flow pathways that extend meters into the soil. In lakes, persistent bubbling churns the water column, revealing the locations of hotspot gas seepage channels in sediments[49], but preferential flow paths in unsaturated, subaerial upland environments are typically invisible at the ground surface, either because the cracks and conduits are too small to see, or because they are obscured by surface soils and vegetation. Detection of high-emission flow pathways in our upland thermokarst-mound sites occurred through the semi-random placement of ~1200 chamber-flux observations. In the future, new ground-based[50] and airborne[51] methods of methane hotspot imaging on land could be useful for improving hotspot quantification in thermokarst-mound environments across the Arctic.

### Geochemistry and microbiology of Yedoma talik soil profiles

To gain a better understanding of subsurface methane dynamics in Yedoma taliks, we conducted borehole soil profile analyses (Supplementary Methods 3.2–3.4). Although NSY is not a wetland, both summer and winter soil geochemical and microbial profile data agreed with the current paradigm for northern wetlands that in situ methanogenesis occurs in deeper, anaerobic soils, and dry surface soils facilitate aerobic methanotrophy[45,52]. NSY borehole sediments sampled in September 2021 showed typical in situ methanogenesis profiles with methane accumulation below 130 cm and peaking around 200 cm (core BH1, Fig. 5a). The measured low $\delta^{13}C_{CH4}$ values were consistent with hydrogenotrophic methanogenesis (approximately −70‰) and coincided with high (heavy) $\delta^{13}C_{CO2}$ values (−15‰). This methanogenic zone also had a high abundance of methanogenic archaea (Fig. S5, Supplementary Data 1 and 3) and elevated expression of the *mcrA* gene, which is responsible for catalyzing the final step of methanogenesis in archaea (Fig. 5a, Supplementary Discussion 5.4).

Above the methanogenic zone, $\delta^{13}C_{CH4}$ profiles provided evidence for the oxidation of dissolved methane with heavier values of the residual methane and lower $\delta^{13}C_{CO2}$ values of the product. We found no evidence of high-affinity methanotrophs (HAM) at NSY, but aerobic methanotrophy seemed to dominate methane oxidation in surface soils, as supported by the relative abundance of low-affinity methanotrophic *Methylobacter* (Gammaproteobacteria; Methylococcales), combined with the methylotrophic bacteria *Methylotenera* (Gammaproteobacteria; Burkholderiales), and by the elevated expression of the *pmoA*

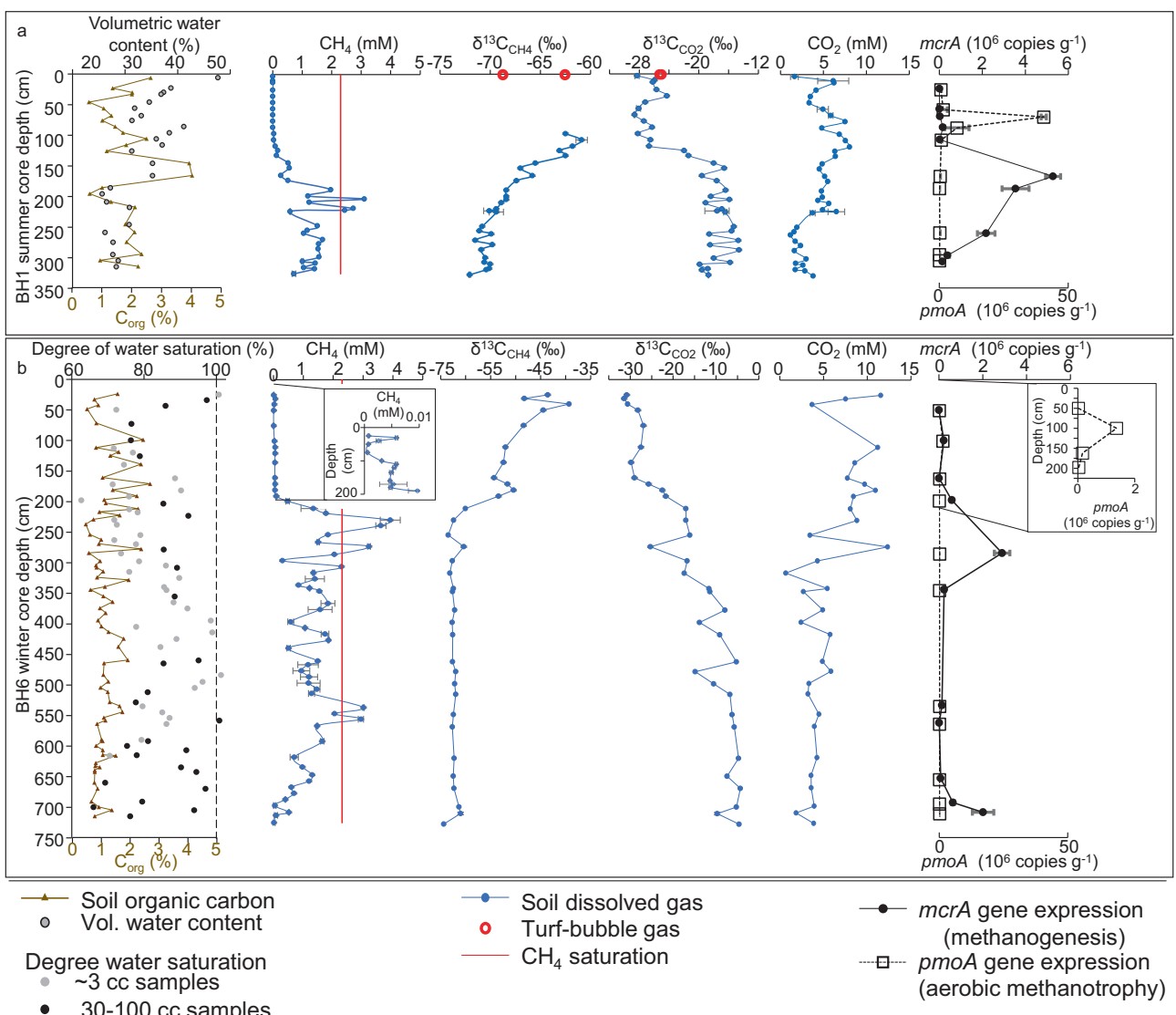

**Fig. 5 | Physical, geochemical, and microbiological properties of the NSY soil profile. a**, summer (BH1) and **b**, winter (BH6) cores. In the winter core, the surface frost layer was ~50 cm thick, and the top of permafrost was observed at 700 cm belowground. Error bars for CH₄ and CO₂ concentrations and their δ¹³C values are the standard deviation (SD) of two technical replicates. Data for *mcrA* and *pmoA* absolute gene expression are presented as mean ± SEM of three technical replicates. Source data are provided as a Source Data file.

gene involved in methane oxidation (Fig. 5, Supplementary Data 1–3, Supplementary Discussion 5.4). In the surface 90-cm of the summer core, methane was absent (Fig. 5a). This indicates that soil-dissolved methane is subject to intensive oxidation near the surface of the silt-dominated soil column in summer, a time when aerobic methanotrophs are active in unfrozen, unsaturated surface soils[20,45,52].

What makes our results particularly unique is that the pattern of soil geochemical and microbial profiles observed at NSY is typical for water-saturated lake sediments, where oxygen-depletion in surface sediments overlies thick anaerobic horizons[53,54]; yet, at NSY, soils were largely unsaturated (Figs. 4 and 5) and not covered by any water column (e.g., thermokarst lake). We suggest that oxygen diffusion into the deep talik is impeded by the low permeability of massive, silt-dominated thawed Yedoma sediments[15] and interstitial water in unsaturated pores. Anaerobic conditions deeper within the talik profile are further facilitated by high respiration in surface soils that consumes atmospheric oxygen and by an increase in soil moisture content, which geophysical and borehole measurements indicate a trend toward saturation at certain intermittent depths (Figs. 4 and 5).

Wintertime freezing of surface soils changed the redox environment of the near-surface soils. Soil oxygen probes indicated development of anaerobic conditions in the surface 15 cm following the onset of seasonal frost, whereas semi-aerated conditions persisted through winter at 50 cm (Fig. S6). This inverted redox pattern may be explained by suction and redistribution of soil moisture to the freezing front from unfrozen soils below[55] (cryosuction) resulting in water(ice)-saturation and anoxia surrounding the 15-cm soil oxygen probe. At the same time, frozen soils are conducive to cracking, which in turn facilitates gas exchange[29,35] that can explain the persistence of some soil oxygen in the underlying, much drier 50-cm soil.

Soil oxygen probes at 100 cm and 190 cm showed anoxia year-round. This was consistent with our observation of no oxygen in the soil porewater samples of a sediment core (BH6) drilled in winter (March 2023) 60-cm away from BH1. Pore spaces of this winter core, which captured the full ~7-m talik profile and underlying permafrost soil down to 7.25 m, were unsaturated, except in the seasonal frost at the ground surface and in several narrow (~10-cm thick) horizons interspersed throughout the lower 3-m of talik. Due to the low hydraulic conductivity of silt, it is possible that these thin, intra-talik water-saturated horizons are moisture-filled voids left from the melting of excess ground ice. Offset from these high-moisture horizons, soil-water-dissolved methane concentrations peaked just below 200 cm and around 550 cm. Elevated abundances of methanogenic archaea and *mcrA* expression in the surrounding, water-unsaturated soil layers coincided with the 200-cm peak (Figs. 5b and S5). However, the shift between the 500-cm methane peak and elevated methanogenic archaea and *mcrA* expression at 700 cm may be due to upward gas migration. Regardless, these results suggest that in situ methanogenesis persisted in the anaerobic talik through winter.

The geochemical and microbial profiles also indicated the likelihood of less methane oxidation in the winter. While soil-dissolved methane levels were below the detection limit in summer above the methanogenesis zone (and therefore also no measurable methane isotopes), in winter, methane was present up to the ground surface, albeit at low concentrations. Wintertime measurable methane concentrations in the upper 200 cm coincided with higher and more variable $\delta^{13}C_{CH4}$ (−51 to −24‰) values compared to summer bubble values ($\delta^{13}C_{CH4}$ −63 to −69‰) and soil-dissolved gas deeper within the talik ($\delta^{13}C_{CH4}$ −69 to −72‰) supporting only partial methane oxidation in the winter with expected larger isotopic fractionation. Suppression of methanotrophy in winter was also supported by the *pmoA* expression, which was 30-fold lower in the winter BH6 core ($1.3 \times 10^6 \pm 3.27 \times 10^5$ copies/1gr soil, at 100 cm depth) compared to the summer BH1 core ($4.05 \times 10^7 \pm 1.02 \times 10^6$; a decrease of 30-fold at 68 cm depth) (Fig. 5b, Supplementary Discussion 5.4). This wintertime

limitation of methanotrophy was likely due to the reduced capacity of oxygen to penetrate ice-filled soil pores in the seasonal frost layer[55] and the metabolic response of methanotrophs to cold temperature[29,55–58]. Anaerobic oxidation of methane seems also to be an insignificant sink of methane, as typical anaerobic methanotrophs were not found. These soil signals indicate that winter inhibition of methanotrophy[20,29,34] allows methane produced in the thick, warmer-at depth[29] (Fig. S6), organic-rich silty talik sediments to release through cracks to the atmosphere.

To further explore the importance of the methanotrophic bio-filter, we experimentally removed surface soils (down to 20–120 cm) at a subset of boreal and arctic thermokarst-mound study sites using a spade in summer and boreholes at different times of the year. In 27 out of 32 measured soil profiles of fluxes over boreholes and soil pits at thermokarst-mound sites across Alaska, we found that surface-soil removal led to an increase in methane emissions and that these emissions increased with depth of soil removal (Fig. 6). Conversely, we observed no increase in methane fluxes when surface soils were removed from seven control (non-thermokarst mound) sites. We do not exclude the possibility that elevated methane would be found at all thermokarst-mound sites, but often tree roots impeded our ability to dig. These soil manipulation experiments point to the effectiveness of aerobic methanotrophy in undisturbed surface soils for mitigating deep-sourced methane emissions from Yedoma taliks. Details of the experiments and investigation of other flux-impeding processes such as seasonal frost (Fig. 6b), the thaw of which led to some of the highest-observed emission rates throughout the year (Table S4a), and winter rain-on-snow events are described in detail in Supplementary Discussion 5.1.

Our soil manipulation experiments also provide evidence that preferential flow pathways (boreholes as an analogy to large cracks) effectively transmit methane to the surface, bypassing oxidation. At several depths in the sediment cores (Fig. 5), talik methane concentrations exceeded saturation level, suggesting the occurrence of free-phase methane. Turf bubble samples also imply free-phase gas migration. Stable isotope values in turf bubbles at NSY were most like those of dissolved gases in the deepest borehole sediments (Fig. 5a), indicating that turf bubble gas originates from the methanogenic zone and follows a different migration pathway than the more isotopically-enriched dissolved gases subject to oxidation as they diffuse slowly through the massive silt soil column[15] (Fig. 7).

### Exceptionally large winter emissions from Yedoma thermokarst-mound uplands

In addition to our extensive plot-scale chamber-flux measurements, eddy-covariance data collected from a tower at mid-elevation in the NSY study field (Fig. 8) revealed the importance of upland Yedoma thermokarst-mound methane and carbon dioxide emissions among terrestrial arctic ecosystems, especially in winter. Chamber-based fluxes were consistent with eddy-covariance fluxes (Fig. S7) observed during the two-year study period characterized by mild temperatures (Table S4a). Total precipitation was higher (450 mm yr$^{-1}$) in the first year (May 10, 2021, through May 13, 2022) than in the second year (231 mm yr$^{-1}$, May 14, 2022, through May 14, 2023; years defined by soil freeze/thaw phenology in "Methods" section), likely contributing to slightly higher methane and carbon dioxide emissions in year 1 (Table S4b). Total annual methane emissions at NSY (45–48 g CH$_4$ m$^{-2}$ yr$^{-1}$, Table S4b) were 240 times higher than northern uplands ($0.2 \pm 0.6$ g CH$_4$ m$^{-2}$ yr$^{-1}$, mean ± SEM, $n = 49$) and nearly three times higher than the mean of 305 global wetlands located north of 60 °N ($16.3 \pm 1.2$ g CH$_4$ m$^{-2}$ yr$^{-1}$)[48] (Fig. 9). NSY carbon dioxide emissions (1141–1254 g CO$_2$ m$^{-2}$ yr$^{-1}$) were approximately twice as high as other terrestrial ecosystems with similar climates[59]. These higher emissions are most likely explained by ≤9-m deep talik formation in massive silty sediments which facilitate anaerobic conditions and preferential gas

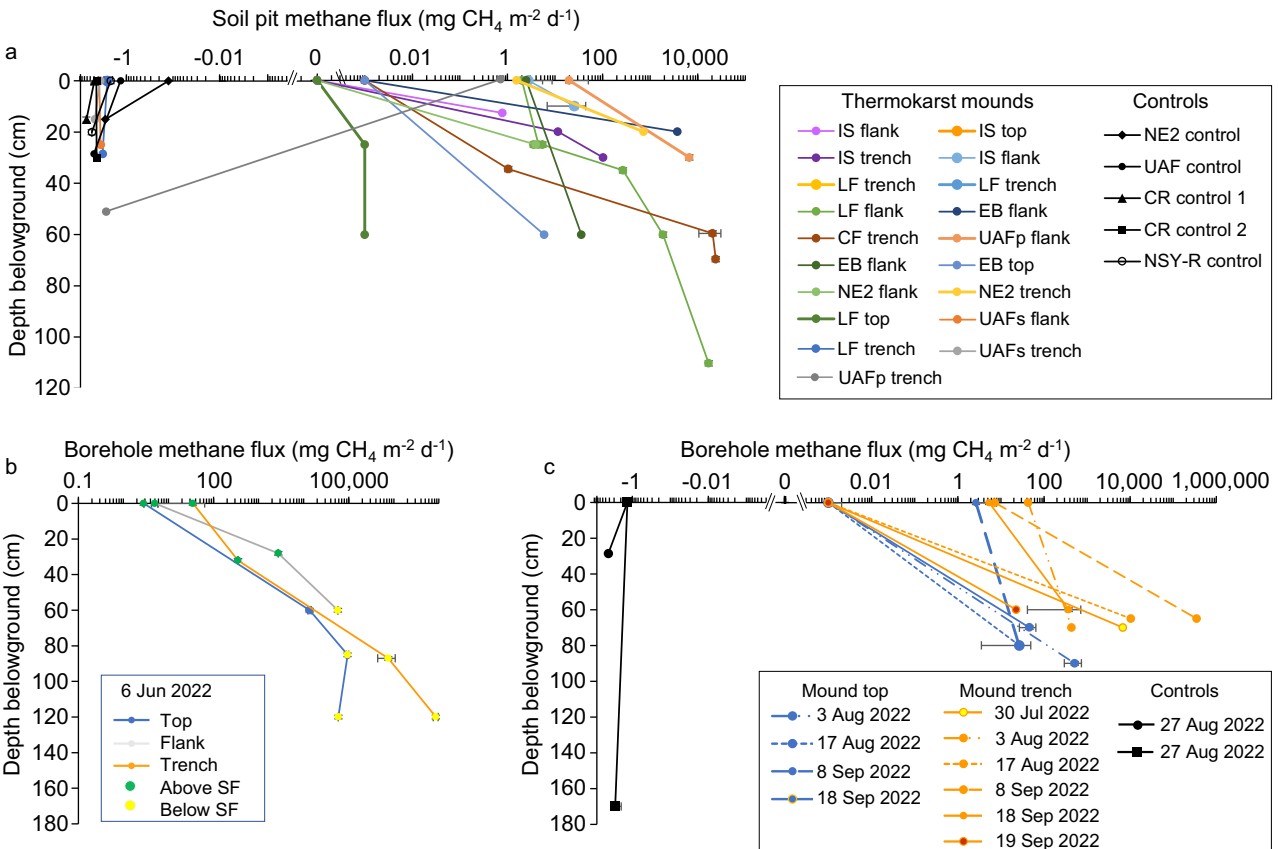

**Fig. 6 | Surface and belowground methane fluxes from soil pits and boreholes.**
**a** Fluxes from ~30-cm diameter soil pits dug with a spade at different thermokarst-mound microtopographical positions (tops, flanks, and trenches) and at control sites in August 2022. See Table S1 for full site names and characteristics. **b** Methane fluxes measured June 6, 2022, at the undisturbed ground surface (depth 0 cm) and various borehole (7.6 cm OD) depths relative to the seasonal frost (SF) at three microtopographical locations on an NSY thermokarst mound. **c** Fluxes measured

over the same boreholes and from boreholes at the UAF black spruce control site later in summer 2022. Among thermokarst-mound soil pits and boreholes, we observed a significant increase in methane flux with depth, and thermokarst-mound profile fluxes were higher than control-site fluxes (Supplementary Discussion 5.1). Source data are provided as a Source Data file. Errors bars are the SD of two or three biological or technical replicates, details of which are provided in the Source Data file.

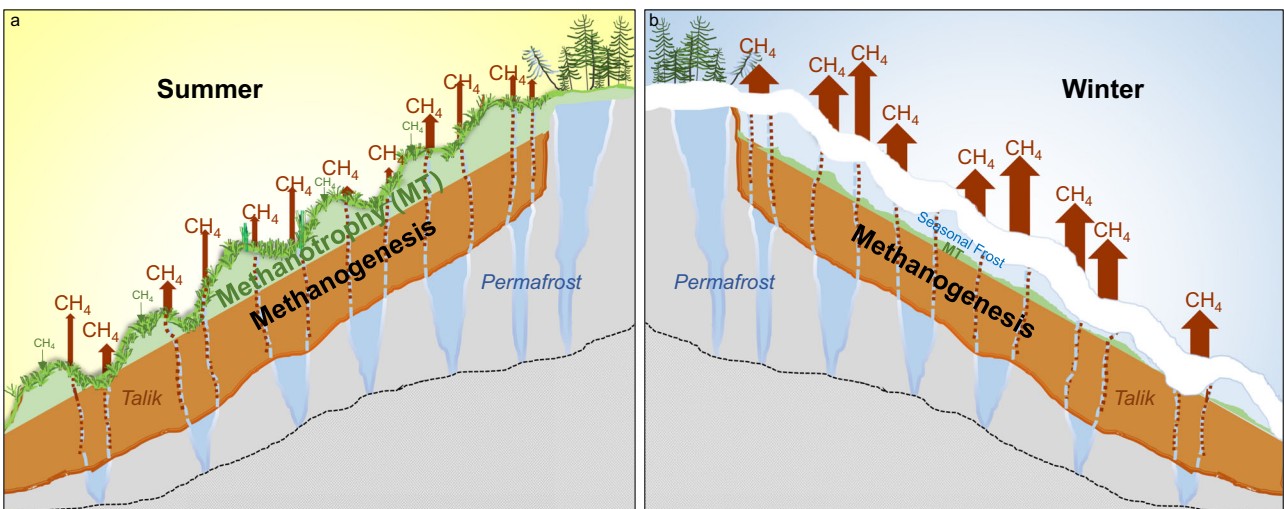

**Fig. 7 | Seasonality of methane (CH₄) dynamics in upland Yedoma thermokarst mounds.** **a** In summer, aerobic methane oxidation dominates surface soils; nonetheless, methane escapes to the atmosphere from anaerobic taliks through preferential flow paths (red dots), such as those formed along melted ice-wedge casts

(red-blue dashed lines in the talik). **b** In winter, relatively larger seasonal methane emissions from the talik are due to a decrease in aerobic methanotrophy by freezing of surface soils. Wintertime methanogenesis in deep talik soils and barometric pressure pumping of methane from taliks also enhance winter emissions.

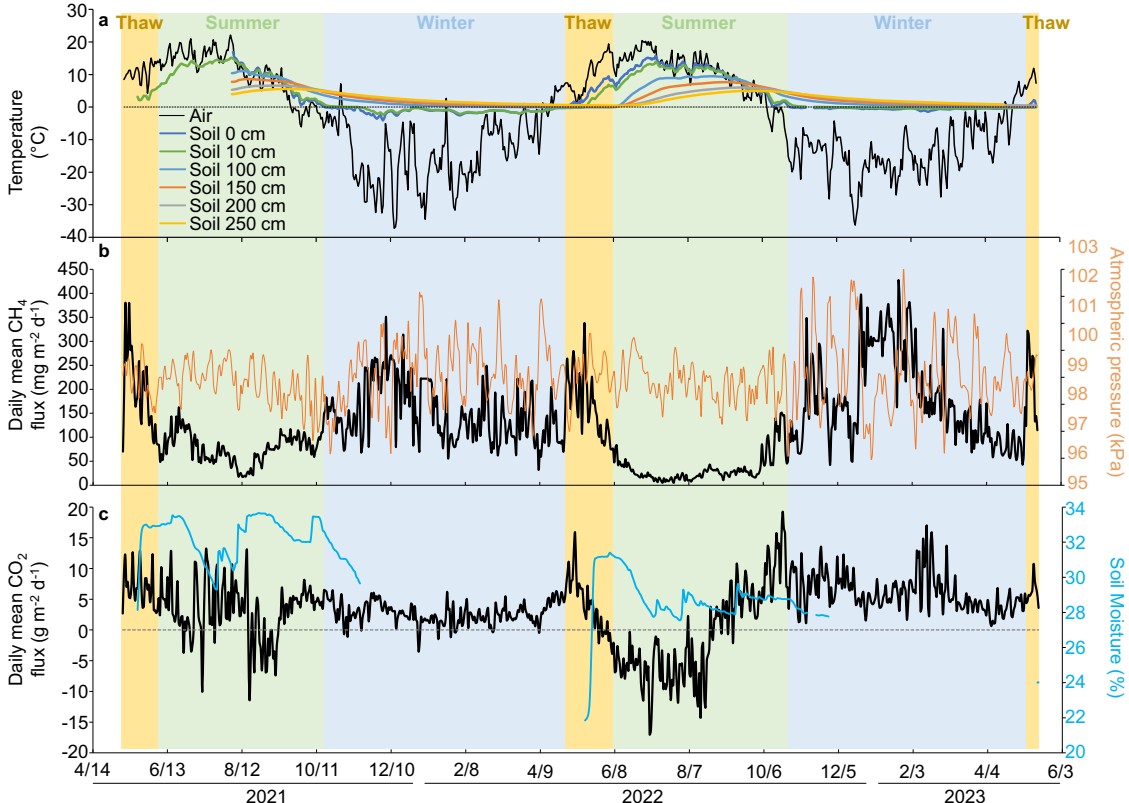

**Fig. 8 | Eddy-covariance tower observations at the NSY thermokarst-mound field.** Air and soil temperature (**a**), methane ($CH_4$) flux and atmospheric pressure (**b**), and carbon dioxide ($CO_2$) flux and soil moisture expressed as volumetric water content at 15 cm depth (**c**).

flow paths in thawing Yedoma soil profiles, environments where carbon stocks are on average five times higher than non-Yedoma upland soils[11,60].

Cold-season methane emissions are increasingly observed in boreal and arctic terrestrial ecosystems[48]. However, these emissions remain relatively small and so far, have been constrained to shoulder-season freeze-up and thaw of the seasonal active layer[29–32]. To our knowledge, seasonal and year-round emissions from upland Yedoma taliks have never before been investigated. Contrary to typical tundra and boreal terrestrial flux tower observations of diminishing methane emissions during winter[29,61,62], we observed an increase at NSY (Fig. 8). Based on our soil geochemical and microbiological analyses, we attribute the high daily winter emissions (165.8 ± 4.0 mg $CH_4$ $m^{-2}$ $d^{-1}$, mean ± SEM, $n = 387$ days), which were about three times higher than summer emissions (55.7 ± 2.3 mg $CH_4$ $m^{-2}$ $d^{-1}$, $n = 274$ days) (Table S4a), to freezing of surface soils, a process that inhibits aerobic methane oxidation and enhances anoxia in the talik (Fig. 7). Other factors that likely elevate thermokarst-mound winter emissions are: (a) the thermal inertia of heat propagation leading to warmest soil temperatures near the base of the talik during early winter (Fig. 8a), boosting winter methanogenesis in taliks[29,63] (Fig. 5b); (b) extreme fluctuations in barometric pressure that pump methane[35] out of taliks in winter (Figs. S8 and S9; Supplementary Discussion 5.2), including methane possibly produced in summer and stored in winter; and conceivably, (c) convection caused by sinking of cold, dense air parcels into cracks, displacing methane from taliks. To our knowledge, the latter mechanism has not been explored in permafrost soils, but air circulation into porous rock beds is most vigorous in interior Alaska during winter[64] and thermal convection is known to enhance gas fluxes within fractures in other dryland soils[65]. At NSY, 62–67% of annual emissions occurred in winter, defined as the 190- to 197-day period from surface soil freeze-up to thaw (Table S4b).

Under the current paradigm, due to their high water table and productive aquatic vegetation, wetlands are thought to emit more methane than upland terrestrial ecosystems on an areal basis[48]. However, we observed the opposite pattern. Winter emissions at our forested and grassland upland thermokarst-mound sites (15.2 ± 4.3, $n = 14$ g $CH_4$ $m^{-2}$ $yr^{-1}$, mean ± SEM) were approximately five times higher than emissions from northern wetlands summarized in Treat et al.[48] (3.2 ± 0.2, $n = 280$ g $CH_4$ $m^{-2}$ $yr^{-1}$) (one-sided Kolmogorov–Smirnov test, $p = 0.002$) (Fig. 9), a difference which we attribute to soil properties. Both wetlands and upland thermokarst mounds have labile carbon substrates, but substrates differ according to source, season, soil depth, and $^{14}$C-age. Within deep taliks of upland thermokarst mounds, methanogenesis continues year-round, fueled by thawing $^{14}$C-depleted permafrost soil organic matter. In contrast, wetland plant productivity, the dominant and $^{14}$C-enriched substrate of most wetland methane[66,67], subsides in winter. Furthermore, gas transmission in winter should be more effective through water-unsaturated soils in Yedoma upland taliks than through water-saturated/ice-bonded wetland soils[32].

## Relevance to permafrost climate feedback

Until recently, subaerial taliks leading to thermokarst-mound formation were predominantly limited to locations where the surface energy balance has been altered by disturbance (e.g. vegetation clearing, road construction, infrastructure development, wildfires, hillslope erosion, and lake drainage)[36,37,40]. In undisturbed areas, icy permafrost soils have historically been protected against degradation by a combination of cool air temperature, low snow accumulation, and insulating surface soil organic layers[68,69]. However, active layer deepening associated with 21st-century climate warming and an increase in winter snowfall[7,8] is projected to alter the surface energy balance, leading to extensive permafrost thaw and talik development[26,70,71]. Recent documentation of widespread thermokarst and novel talik development in

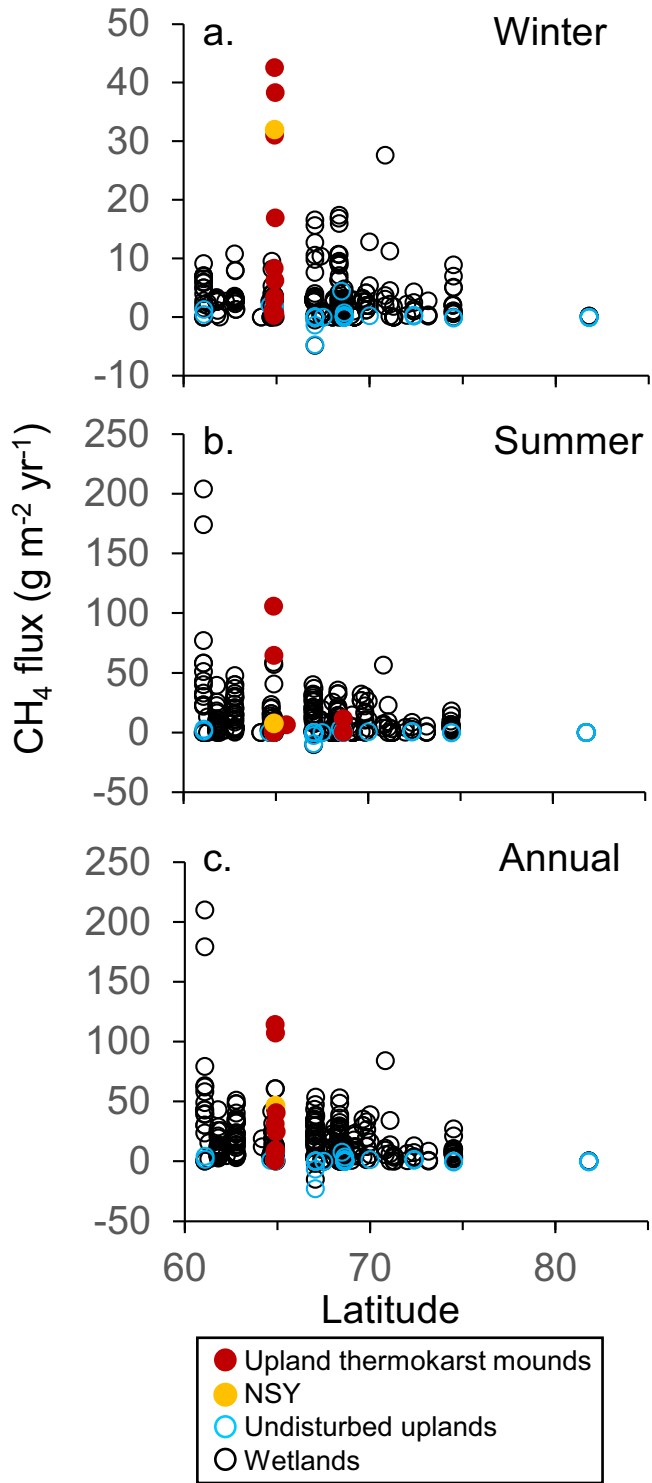

**Fig. 9 | Ecosystem methane emissions by latitude. a**, summer; **b**, winter, and **c**, annual. Northern wetlands ($n = 305$) and uplands ($n = 49$) are from Treat et al.[48].

Legend:
- ● Upland thermokarst mounds
- ● NSY
- ○ Undisturbed uplands
- ○ Wetlands

rate[10]. By 2090 taliks beneath black spruce forests, the second coldest ecotype in the discontinuous permafrost zone, may reach a thickness of 12 m[9].

To assess the potential for thermokarst-mound and talik formation in well-drained upland Yedoma deposits (i.e., the remnant of Pleistocene-aged Yedoma that has not yet been degraded by thermokarst or other erosional processes during the Holocene), we simulated the geomorphological and hydrothermal evolution of thermokarst-mound taliks under the unmitigated representative concentration pathway (RCP) 8.5 strong climate-warming scenario ("Methods" section) (Fig. S10; Tables S5 and S6) following the tile-based modeling approach of Nitzbon et al.[74]. As a contrast to the relatively warm climate and discontinuous permafrost conditions of our interior Alaska field study sites, we selected for our modeling the central Lena River Delta in northeast Siberia, belonging presently to the coldest part of the Yedoma domain[2]. Our simulations consistently indicated ground subsidence and thermokarst-mound development followed by talik formation in the 21st century extending into the 22nd century (Figs. 10 and S11; Supplementary Movies 1–6). Variations of thermokarst-mound geometry, initial topography, and drainage efficiency showed modest effects on the overall thaw trajectory. Meanwhile, model parameter variations that enhance snowpack insulation within plausible ranges (increased snowfall by 50%; decreased snow density by 20%) resulted in up to several decades earlier and up to 10 m deeper talik development (Fig. 10). This demonstrates the vulnerability of presently-cold, ice-rich permafrost landscapes to expected increases in winter precipitation[7,8]. Simulations further revealed that laterally transported heat from the relatively warm trenches and mound flanks amplifies talik formation, and thus potentially spurs microbial mineralization deep inside thermokarst mounds several years to decades earlier than expected under gradual thaw conditions (Fig. 10b).

This modeling experiment, together with our remote sensing and fieldwork observations of widespread thermokarst-mound occurrence in the Yedoma domain already today (Fig. 1), indicates that taliks are highly likely to develop across the majority of the upland Yedoma landscape during the 21st and 22nd centuries in a strong climate-warming scenario. Additional mounds may form in non-Yedoma polygonal systems under the precondition of the presence of large ice wedges (e.g. sites E6 and NE2 in this study); however, these mounds will typically be smaller than Yedoma thermokarst mounds owing to smaller ice-wedge dimensions. A fraction (≤29%) of Yedoma domain thermokarst-mound taliks are also expected to form beneath Yedoma thermokarst lakes[75], but these subaqueous environments are already known for high methane emissions[16]. What is critical to our thinking about permafrost carbon feedback is the potential for upland Yedoma thermokarst-mound taliks to produce and emit methane. Contrary to existing model projections of carbon-dioxide-driven radiative forcing associated with upland permafrost thaw[6,10,14,18,26], our work shows that where subaerial taliks form in ice-rich Yedoma landscapes, methane contributes approximately ten percent of total emissions on a carbon-mass basis, but fifty-two percent of the climate forcing (Table S4b). These observed Yedoma upland methane emissions constitute a -9 to 57-fold increase over current model predictions[14,18,26].

Our study brings to light complexities that change the way we should think about permafrost carbon feedback in the Yedoma domain. Current assumptions that terrestrial uplands will remain an insignificant source of methane or become a stronger sink[14,18,22,23,26] will likely lead to considerable underestimation of the radiative forcing associated with greenhouse gas fluxes in upland Yedoma taliks. Upland Yedoma landscapes with thermokarst mounds differ from other upland environments owing to the development of anaerobic taliks in largely unsaturated, but thick (tens of meters) organic-rich silts. This, together with the formation of post-cryogenic structures that facilitate methane transport from meters belowground, bypassing oxidation, cause Yedoma uplands with thermokarst mounds to have some of the

undisturbed uplands of Siberia[72], Canada[73], and Alaska[9,68] suggests an important climate threshold has already been crossed. Exceptionally warm air temperatures and high snowfall in winter 2018 led to the initiation of subaerial taliks in 44% of the undisturbed discontinuous permafrost sites monitored across Alaska[9]. These observations are consistent with model projections that widespread talik development will likely accelerate early in the 21st century and peak after 2050, affecting 14.5 million km² if climate warming continues at the current

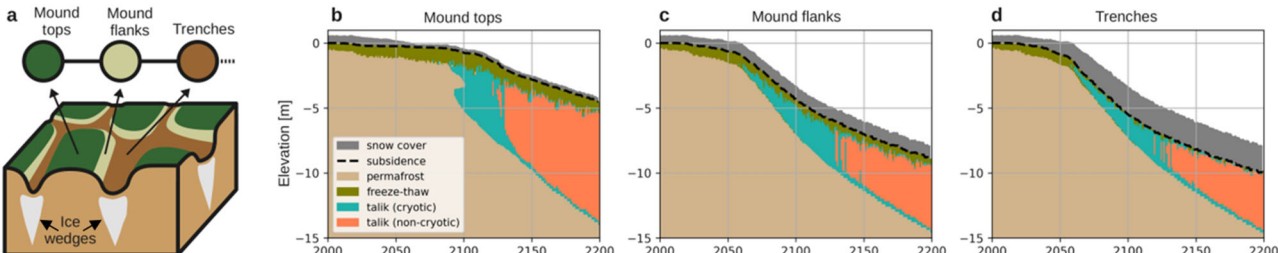

**Fig. 10 | Simulated thermokarst-mound and talik formation for the Lena River delta in Northeastern Siberia under RCP8.5.** Schematic of the tile-based abstraction of an upland thermokarst-mound landscape (**a**), and modeled thaw subsidence (dashed line), maximum snow heights (gray area), and evolution of the hydrothermal state (colored areas) for mound tops (**b**), mound flanks (**c**), and trenches (**d**). Note that cryotic talik formation (minimum annual soil liquid water content >10% and perennially unfrozen soil temperature <0 °C) due to laterally transported heat precedes the development of a non-cryotic talik (perennially unfrozen soil temperature >0 °C) underneath mound tops by several decades. Shown results are for a simulation with a snowfall multiplier of $\varphi = 1.0$ and a snow density of $\theta = 200$ kg m$^{-3}$. Simulation results for further parameter settings are shown in Fig. S11 and Supplementary Movies 1–6. Panel a is adapted from "Effects of multi-scale heterogeneity on the simulated evolution of ice-rich permafrost low-lands under a warming climate" by Nitzbon, J. et al. [92], published under CC BY 4.0.

highest methane emissions yet documented among terrestrial Arctic ecosystems, especially in winter.

Extrapolation to larger spatial extents will require further examination of how representative our interior Alaska findings are for other regions, particularly with respect to talik development rates and corresponding carbon mineralization, controls over methanotrophy, and permafrost hydrology. However, upland taliks are expected to spatially dominate the 21st- and 22nd-century arctic landscape[10], and our work shows that the entire Yedoma domain is susceptible to thermokarst-mound formation. Given the disproportionately large concentration of the permafrost soil organic carbon pool in Yedoma (~25%) and the exceedingly high methane emissions observed in Yedoma thermokarst mounds, representation of this important landform among other critical methane-emitting ecotypes (i.e. thermokarst wetlands and lakes) will likely improve permafrost carbon feedback modeling[10,14,18,26].

## Methods

Methods utilized in this study include (a) collection of turf bubble samples for gas composition and isotope analyses, (b) eddy-covariance measurements of methane ($CH_4$) and carbon dioxide ($CO_2$) fluxes at North Star Yedoma (NSY), (c) plot-scale measurements of the same gas fluxes using a portable chamber system at NSY and 25 other extensive thermokarst-mound study sites in Alaska, (d) geophysical measurements of soil properties, (e) borehole soil analyses, (f) remote sensing detection of thermokarst-mound occurrences in the pan-Arctic, and (g) numerical modeling of talik development in Northern Siberia (for d–g, see Supplementary Information 3). Soil analyses (e) included determination of moisture content, dry density; organic carbon and nitrogen concentrations; soil-dissolved $CH_4$ and $CO_2$ concentrations; $\delta^{13}C$ values of $CH_4$ and $CO_2$; qPCR analysis of the *mcrA* and *pmoA* genes, and 16S amplicon-based sequencing for characterization of the microbial community. Statistical approaches are also described.

## Study sites

The NSY study site is located seven kilometers NW of Fairbanks, Alaska. From 1970 to 2022 this subarctic, continental region had a mean annual air temperature of $-2.2 \pm 1.2$ °C and mean annual precipitation of $288 \pm 75$ mm water equivalent[76]. NSY was chosen for eddy-covariance flux monitoring and other intensive measurements because it is a large open field that meets the requirements of eddy-flux[77]; has electrical grid access; and has thermokarst features, vegetation and hydrology similar to those of our extensive study sites where thaw initiated in the last 40–70 years (Table S1a). NSY is part of a ~60-hectare field of thermokarst mounds formed following anthropogenic disturbance to the mature black spruce forest. Landsat imagery indicated the initial disturbance occurred sometime between August 1976 and September 1978. Vegetation and surface organic soils were cleared with a bulldozer and then burned. Following this initial disturbance, natural vegetative succession accompanied by intense thermokarst-mound development occurred until 2000, when mounds were leveled in the eastern half of our study field, and then disced prior to seeding turf grasses for the establishment of a rugged golf course. The western half was untouched, allowing for higher-relief mound development and natural forest succession. After 2000, ground-ice melt and thermokarst subsidence continued across the whole field resulting in the mound-ridden surface at NSY today. The gradually-sloping field has beaded drainage[37], consisting of several ephemerally flowing water tracks along S to N and SE to NW directions with tiny (~1 m² to 79 m²) ponds at the junctions of ice wedges. Analysis of high-resolution optical satellite imagery indicated that the ponds appeared sometime between 2003 and 2011. On May 23, 2021, during the high-water season following snowmelt, the total area of ponds with associated littoral vegetation and water-filled trenches was 391 m² or 2.4% of 16,020 m² that is within the EC tower footprint 80% of the time (Fig. 2). During the drier summer of 2022 water levels in the ponds dropped and some ponds were entirely dry. Despite occasional re-discing and re-grading with small bulldozers, mound formation at NSY progressed. No foreign sediments were ever introduced to the site and no fertilizer was applied during our study period.

The thermokarst-mound field is dominated by two native grasses, *Calamagrostis lapponica* and *Calamogrostis canadensis*. Grasses are also found in thermokarst-mound environments in Siberia where humans have had no impact[39,40]. From May 8, 2021, the start of eddy-covariance tower data collection, until May 2022, the golf course was non-operational and grasses grew with minimal mowing disturbance. In May 2022, following the turnover of property ownership, the golf course resumed operation, and areas outside the 60% footprint were mowed weekly in summer. This grass mowing in 2022 decreased gross primary productivity, providing less of an offset to carbon dioxide emissions (Fig. S12).

Vegetation on the higher-relief thermokarst mounds in the regrowth area, which was never operated as a golf course, was dominated by paper birch (*Betula neoalaskana*) and willows (*Salix* spp.) with occasional white spruce (*Picea glauca*). Ground cover in the regrowth area was mostly leaf litter, with some common feather moss (*Hylocomium* spp.), *Dicranum* spp., and liverworts. *Sphagnum*, a late succession species, was absent from both the mowed and regrowth areas of NSY.

The mature black spruce (*Picea mariana*) forest immediately south and east of our study site was used as a control. Depth to

permafrost (i.e., active layer depth) in the mature black spruce forest was 67 cm on Sep. 15, 2021. Borehole data within 800 m of NSY indicate that organic-rich silts extend 46 m belowground and the total depth of permafrost is ~40 m[78]. Understory vegetation consisted of mosses (*Hylocomium* spp., *Pleurozium schreberi*), Labrador tea (*Rhododendron groenlandicum*), wood horsetail (*Equisetum sylvaticum*), petite dogwood (*Cornus canadensis*), lowbush cranberry (*Vaccinium vitis-idaea*), rosehip (*Rosa acicularis*), and occasional mounds of *Sphagnum* spp. moss. Details of the extensive upland thermokarst-mound study sites in interior Alaska and the north slope of the Brooks Range (Fig. S1), including their locations, vegetation, and disturbance history are provided in Table S1a. Control sites were undisturbed sites, paired with adjacently disturbed thermokarst-mound sites in similar ecological and geologic settings.

## Turf bubble sampling

Two sets of turf bubble samples were collected from NSY on September 13, 2018, by piercing with a knife the dense, surface organic horizon of the soil profile at the toe slopes of two thermokarst mounds. Free-phase gases trapped beneath the turf escaped through the holes into a water-filled inverted funnel set atop the grass. Pressure was applied to the funnel to create a hermetic seal against the turf. The funnel consisted of a sawed-in-half, capped, 2-L plastic bottle. Bubbles entering the bottle displaced the water and accumulated at the top, under the bottle's cap. The cap was outfitted with a tube and stopcock, enabling syringe sampling of the bubbles and transfer to glass serum bottles without contamination of atmospheric air. A photograph of the sampling and video of NSY turf bubbles can be found at KTOO[42]. In addition to turf bubbles, we used submerged bubble traps and glass serum bottles to collect ebullition bubbles from two bubbling point sources in a small thermokarst pond formed in an NSY water track. Turf bubble gas concentrations of $CH_4$, $CO_2$, $O_2$, and $N_2$ were measured by gas chromatography at Florida State University; additional samples were measured at the Colorado Plateau Stable Isotope Laboratory. Stable isotope values, $\delta^{13}C_{CH_4}$ were determined using a Finnegan Mat Delta V (Thermo Electron, Waltham, MA) at Florida State University. Stable isotope compositions are expressed in $\delta$ (‰) $= 10^3$ (($R_{sample}/R_{standard}$) − 1), where $R$ is $^{13}C/^{12}C$ and the standard refers to the Vienna Pee Dee Belemnite (VPDB). The analytical error of the stable isotopic analysis was ±0.1‰ $\delta^{13}C$. $\delta^{13}C$ on pond bubble samples was measured at the University of California Irvine (UCI) via isotope ratio mass spectrometry (GasBench II, DeltaPlus XL, Thermo Fisher Scientific, Pittsburgh, PA, USA). Pond bubble $CH_4$ and $CO_2$ concentrations were determined along with radiocarbon ages of $CH_4$ and $CO_2$ in the turf and pond bubble gas at the W. M. Keck Carbon Cycle AMS (KCCAMS) Laboratory at UCI. Methods described by Elder et al.[79] were followed, including flow-through vacuum extraction to separate and purify $CH_4$ and $CO_2$ from bulk gas, chemical reduction to graphite, and $^{14}C$ analysis via accelerator mass spectrometry (AMS)[80].

## Eddy covariance (EC)

Carbon (net ecosystem exchange of $CO_2$ and $CH_4$ flux) and meteorological data were collected year-round from May 8, 2021, to May 14, 2023, with eddy-covariance measurements. This eddy-covariance system was mounted at 2 m height on a tripod in the center of the NSY grassland study field (64.894 °N, 147.637 °W). Footprint analyses[81] indicate that at least 80% of the flux contribution came from the grassland area within the target thermokarst-mound ecosystem (Fig. 2c). The remainder included a portion of the reforested thermokarst mounds. Within the 80% flux contribution area (Fig. 2c), thermokarst ponds and their associated littoral vegetation comprised a maximum of 2.4% of the area and 7.9% (0.8%) of the $CH_4$ source observed by the EC tower in summer (winter).

Until November 2022, when line power was established at NSY, electrical power for instrumentation was provided by a single 250 W

solar panel and twelve 6-volt absorbent glass mat batteries, with the weekly use of a generator for ~12-h battery charging during the months of December and January. Flux data collected during generator run time were discarded.

The eddy-covariance instrumentation included a 3-D sonic anemometer (CSAT-3B; Campbell Scientific Instruments, Logan, Utah, USA), an open-path infrared gas analyzer for $CO_2$, water, and energy fluxes (LI-7500DS IRGA; LI-COR, Lincoln, Nebraska, USA), and a fast-response open-path $CH_4$ analyzer (LI-7700; LI-COR, Lincoln, Nebraska, USA). The main axis of the LI-7500DS IRGA was tilted by 30° with respect to the horizontal to aid in draining condensation and precipitation from the optical windows. The sonic anemometer and both IRGAs were mounted on a shared horizontal bar and were laterally separated by <20 cm to reduce flux loss and flow distortion. All fast-response instrumentation was connected to a SmartFlux3 (SF) unit (LI-COR, Lincoln, Nebraska, USA) to log raw data at 10 Hz. Raw 10 Hz and 30 min mean data were stored on the SF USB drive. The gas analyzers were calibrated every 3–4 months following the instructions in the instrument manuals [LI-COR Inc., 2020] since inspections indicated that the instruments remained stable over that time frame.

Basic microclimatic data were also collected, including air temperature (Ta) and relative humidity (RH; 2 m above the ground; EE181, Campbell Scientific Instruments), soil water content (VWC; at 15 cm depth, CS616, Campbell Scientific Instruments), precipitation (P; at 0.2 m; TE525MM, Texas Electronics, Dallas, Texas, USA), short and longwave radiation components, net radiation and albedo (SWin/out, LWin/out Rn, albedo; at 2 m above the ground; NR-01; Hukseflux, Delft, Netherlands), soil temperature (Ts at 1 and 10 cm depth; 107 thermistor probe; Campbell Scientific Instruments), and barometric pressure (Pa; Licor LI-7500DS). These variables were measured at 1 Hz and stored on the datalogging system (CR1000X; Campbell Scientific Instruments). Both the processed eddy-covariance and microclimatic data were averaged for 30 min periods.

## EC data processing and post-processing

Eddy-covariance flux calculations were made using Matlab R2017b and EddyPro 7.0.8 software (LI-COR Env., Lincoln, NE); flux data post-processing was done using ReddyProc R package as described in Euskirchen et al.[62,82], which we summarize here. A $CO_2$ signal strength diagnostic, which represents optical impedance by precipitation or aerial contaminants is provided by the infrared gas analyzers. This diagnostic was used as a quality assurance/quality control variable for both flux and radiation data. The WPL terms were applied during post-processing to the $CO_2$ and latent heat fluxes to account for changes in mass flow caused by changes in air density[83]. Corrections were applied to account for frequency attenuation of the eddy-covariance fluxes[84,85]. To account for nocturnal $CO_2$ advection, we calculated a storage term and then performed a friction velocity ($u*$) correction for calm periods, when $u*$ was less than 0.08 m s$^{-1}$.

Raw flux data acquisition during the entire period was about 67% after accounting for data loss from power outages, instrument malfunction, precipitation, and initial flux processing filtering. Data post-processing such as $u*$ filtering and quality flagging further reduced data coverage to about 54%. Data gaps occurred because of instrument malfunction, power outages, or occasional generator use in December and January. Shorter gaps in the eddy-covariance data were usually related to instrument errors during precipitation events in the summer and winter. Longer gaps occurred due to power outages and instrument shutdowns during cold temperatures. For data gaps in net ecosystem exchange (NEE) and $CH_4$ of approximately 1–6 days, we gap filled by calculating the mean diurnal variation, where a missing observation is replaced by the mean for that time period (half hour) based on adjacent days[86]. This method provided stable approximations of missing data using 7-day independent windows during the nighttime hours and 14-day windows during the daytime hours[86].

In addition to gap-filling, we also calculated a daily mean flux. If a day had more than 5 half-hour period observations, a mean was taken for that day. That mean (in $\mu$mol m$^{-2}$ s$^{-1}$) was converted to mg $CH_4$ m$^{-2}$ s$^{-1}$ (for a 30 min period) and multiplied by 48 since there are 48 half-hour periods in a day. Gap-filled data are presented in this study, but the daily mean approach yielded similar results.

We did not directly measure gross primary productivity (GPP) and ecosystem respiration (ER) with eddy covariance, but the NEE based on eddy-covariance data was partitioned into these counterparts to provide an approximation of ER and GPP and therefore a general understanding of the photosynthetic versus respiratory controls over NEE. This partitioning was calculated by employing the algorithm described in Reichstein et al.[87], using the ReddyProc software[87–89]. The partitioning was performed based on nighttime temperature, where nighttime was defined as PAR < 50 $\mu$mol m$^{-2}$ s$^{-1}$. The algorithm fit a respiration model to the measured nighttime NEE data and then extrapolated the optimized model to the daytime using temperature observations during the day. An adaptive model extrapolating ER response to temperature was developed to estimate ER over time. The difference between modeled ER and measured NEE provides the GPP estimate.

## Soil temperature and oxygen

The NSY soil temperature profile was measured with thermistors adjacent to the eddy-covariance tower on a thermokarst mound. Ground temperatures at four depths per profile down to 2.5 m below the ground surface were recorded at one-hour intervals by a HOBO datalogger (4-channel UX120-006) wired to TMCx-HD thermistor temperature sensors (Onset Computer Corp; accuracy 0.10 °C). Prior to installation, we performed an ice bath calibration to improve the accuracy of temperatures near 0 °C to within ±0.06 °C. Hence, zero curtain temperature windows and dates were considered 0 ± 0.06 °C. Snowpack disturbance was minimized during winter.

We delineated the seasonality of fluxes based on daily mean temperatures along the soil profile. Winter was defined as the period when the ground surface temperature was <0 °C. The thaw period commenced when the surface soil temperature rose above 0 °C and ended when the 1-m soil depth rose above 0 °C (Fig. 8). Summer was the period when the entire soil profile was thawed above the base of the talik. We measured soil oxygen concentrations using Apogee SO-411 sensors. Two sensors were initially installed in the thermokarst-mound flank adjacent to the EC tower at 10 cm and 100 cm soil depths on June 6, 2022. The shallow sensor was moved to 15 cm depth on June 24, 2022, and the deep sensor to 50 cm on August 4, 2022. Two additional sensors were installed next to the tower on the mound top at 100 cm and 190 cm depth on May 23, 2023.

## Chamber-flux measurements

Plot-scale $CH_4$ and carbon dioxide fluxes were monitored at NSY and other extensive thermokarst-mound and control sites in Alaska (Table S1) using portable chambers following the methods of Elder et al.[51], which we describe here. At NSY we performed repeat chamber-flux measurements along sections of E−W and N−S transects shown in Figs. 2c and S2a as well as along a short ~50 m N−S transect crossing several thermokarst mounds adjacent to the tower. Less frequent measurements were made at numerous other locations across the study field, in the regrowth forested area with higher-relief thermokarst mounds and in the undisturbed old-growth black spruce forest control sites. Chambers (660 cm$^2$) consisted of opaque, plastic 20-L buckets with bottoms removed and resealable air-tight lids (Gamma Seal Lid, Encore Plastics, Sandusky, OH, USA). Chambers blocked 88% (41%) of photosynthetically active radiation measured with a cosign quantum flux sensor (Apogee Instruments, Logan, UT) in full sun (shade). Chambers were placed on the ground surface, snow surface, or semi-permanent collars for individual flux measurements. Air was recirculated through the chamber and either a Los Gatos Research Ultra-Portable Greenhouse Gas Analyzer (UGGA) or Los Gatos Research Micro-Portable Greenhouse Gas Analyzer (MGGA) (ABB INC., Quebec City, CA) with a $CH_4$ concentration measurement frequency of 1 Hz. Diffusive fluxes were calculated from the ideal gas law using chamber volume, temperature, and atmospheric pressure measured via the eddy-covariance tower's LI-COR LI-7700 (LI-COR Inc., Lincoln, Nebraska, USA) operating mid-field at NSY. The slope of linear $CH_4$ concentration change (usually $R^2 > 0.90$ correlation to linear least squares fit) for a minimum of 45 s (45 observations) and a maximum of 210 s ($45 < n > 210$) was used to determine a mass change within the chambers. Of 966 linear-flux measurements, 92% had an $R^2$ between 0.90 and 0.99; 8% had an R$^2$ between 0.8 and 0.9. An additional 291 observations of zero (neutral) flux were included based on no change in concentration within chambers. Chamber observation periods were ≤5 min. Seventy-nine observations with non-linear concentration change ($R^2$ of linear fit <0.80), no linear sections ≥45 s, or stepwise concentration increases (interpreted as ebullition) were omitted to ensure that the reported measurements represent purely diffusive fluxes. These strict protocols eliminated any disturbance caused by placing the chamber, which we assume would manifest as a non-linear $CH_4$ concentration change. See Elder et al.[51] for more details regarding the processing of raw chamber data and data quality protocols.

All fluxes were measured between the hours of 09:00–21:00 local time. Chamber-based fluxes were not measured at night; therefore, we assume our daytime measurements capture the diurnal variability of $CH_4$ emissions. This may represent a source of error in our extrapolations, though this potential effect was not quantified. During periods when surface soils were thawed, we measured volumetric water content at 12 cm soil depth at the locations of our chamber fluxes using an HS2 HydroSense soil moisture sensor (Campbell Scientific, Logan, UT, USA). An additional 40 measurements at NSY were made over shallow open water or emergent vegetation along the margins of small thermokarst ponds forming in the NSY water tracks. These data were excluded from our chamber-based assessment of thermokarst-mound fluxes; however, they were considered in calculations assessing the potential contribution of ponds to the tower fluxes. Plant-mediated fluxes when vegetation was present in the chamber were linear. Thus, fluxes reported herein should be considered as inclusive of linear plant-mediated emission exclusive of ebullitive emissions. Ebullition emissions from open water pond areas were estimated by applying mean-daily ebullition observed through year-round bubble-trap monitoring in Yedoma lakes[49] to the number of high-flux seeps observed in the NSY ponds (Supplementary Discussion 5.4).

## Statistics

Prior to using analysis of variance (ANOVA), we checked that our data [untransformed, log-transformed, or -rank-transformed], met standard assumptions, and tests were run using an alpha level of 0.05 unless otherwise specified. If data did not meet the assumptions, we used alternative, non-parametric statistical tests.

**Chamber-flux data.** We performed a two-way Type III ANOVA on rank-transformed data to analyze the effect of the study site and type (thermokarst mound vs control) on plot-scale chamber-based $CH_4$ fluxes. The same test was used on rank-transformed data to test the effects of thermokarst-mound site and season on plot-scale $CH_4$ fluxes. We used Pearson's product-moment correlation test on log-transformed data to test the relationship between surface soil moisture and $CH_4$ flux at NSY and other thermokarst-mound sites. We used the Kruskal−Wallis rank sum test followed by the Wilcoxon pairwise comparisons to determine differences in chamber-based $CH_4$ fluxes according to microtopographical position at the extensive thermokarst mound sites and NSY.

**Eddy-covariance data.** We used the Kruskal–Wallis rank sum test to analyze differences in eddy-covariance $CH_4$ and $CO_2$ fluxes by study year. We used the same test followed by the Wilcoxon rank sum pairwise comparison test with continuity correction to assess differences in fluxes by season within different study years. We used linear regression to analyze the relationship between changes in barometric pressure and $CH_4$ flux for a selection of 30 pressure events across summer and winter seasons and for the full study period. We used one-way ANOVA followed by linear regression to test the relationship between cumulative barometric pressure deviation and direction of deviation (positive or negative) on the cumulative deviation in $CH_4$ flux for time-integrated observations from January 11, 2022, through May 3, 2022. We used the one-sided Kolmogorov–Smirnov test for differences in winter $CH_4$ emissions between thermokarst mounds and northern wetlands.

**Manipulation-experiments data.** We used the Wilcoxon signed rank test on paired differences to test if chamber $CH_4$ flux was related to the removal or intactness of snow-ice layers from a mid-winter rain event. To test the relationship between chamber measurements of $CH_4$ fluxes from above and beneath seasonal frost layers we used the Kruskal–Wallis rank sum test. We used type III two-way ANOVA on log-transformed $CH_4$ flux data to test for the effects of microtopographical position and borehole depth on June 6, 2022 $CH_4$ fluxes. For borehole measurements performed later throughout the summer of 2022, we used type III two-way ANOVA on $CH_4$ measurements from boreholes, testing $CH_4$ flux response to site type (thermokarst mounds vs. controls), borehole depth, and type x depth interaction. Methane observations were rank-transformed to normalize residuals. We performed the same analysis on the soil pit data set. We also performed type III two-way ANOVA on $CH_4$ measurements from thermokarst-mound boreholes alone, testing $CH_4$ flux response to borehole site, borehole depth, and site x depth interaction. Methane observations were rank-transformed to normalize residuals. The same analysis was performed on soil pit $CH_4$ fluxes among thermokarst mounds. Finally, we used the Kruskal–Wallis rank sum test to determine if soil and vegetation management impacted $CH_4$ flux at the Isabella thermokarst-mound site.

**Geophysics.** We used an F-test to identify which spectral peaks in our analysis of the spatial periodicity of geophysically derived water content were statistically significant with 90% confidence.

Statistical analyses were performed in R version 4.2.2, using packages car, ggplot2, MASS, and multcompView.

### Reporting summary
Further information on research design is available in the Nature Portfolio Reporting Summary linked to this article.

## Data availability
The raw sequence reads generated in this study have been deposited in the European Nucleotide Archive (ENA) at the EMBL European Bioinformatics Institute (EMBL-EBI) Database (https://www.ebi.ac.uk/ena/browser/home) as BioProject accession number PRJEB59938 (https://www.ebi.ac.uk/ena/browser/view/PRJEB59938). Eddy-covariance data generated in this study have been deposited at AmeriFlux and are available as site US-YNS (North Star Yedoma) under accession code https://ameriflux.lbl.gov/sites/siteinfo/US-YNS. Eddy-covariance daily mean fluxes, soil and air temperature and soil moisture data in this study have been deposited in the Arctic Data Center under accession code https://doi.org/10.18739/A22V2CC2M. Chamber-based methane flux data generated in this study have been deposited in the Arctic Data Center under accession code https://doi.org/10.18739/A26W96B49. Geophysical data generated in this study have been deposited in the U.S. Geological Survey database under accession code https://doi.org/

10.5066/P9XEMDE1. Additional data are available in the Supplementary Information and Supplementary Data sections. Source data are provided with this paper.

## Code availability
The datasets and scripts that were used for the preparation of Fig. S5b, c have been uploaded to the GitHub repository at: https://github.com/BergmanOded/Yedoma_Nat.Commun2024, digital resource identifier: https://doi.org/10.5281/zenodo.11561649, Bergman[90]. The model code and set-up files for the numerical modeling with CryoGrid are available from https://doi.org/10.5281/zenodo.3648266.

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

## Acknowledgements

Site history and access to the North Star Yedoma field site were provided by Roger and Melinda Evens and Raymond and Stephanie Nadon. Numerous other property owners granted access to land for portable chamber-flux measurements. Chuck Smallwood, Bryce Ricken, Monica Sanchez, Stephanie Kolker, Benjamin Jones, and Maria Teresa Aguirre-zabala Compano assisted with fieldwork. Prajna Lindgren assisted with historical image analysis. Ben Jones provided UAV imagery in Figs. 3f and S1c. Funding was provided by NSF AON 1936752 (K.M.W.A., C.E., A.K., S.Z., E.E., M.S.B.-H.), NSF NNA 2022561 (K.M.W.A., P.A.), NASA ABoVE (K.M.W.A., N.H.), Sandia National Laboratories (K.M.W.A., N.H.), USGS National Land Imaging program (B.J.M., S.R.J., N.J.P.), ERC 818450 and ISF 1573–2022 (O.S., E.E.R., O.B.), European Space Agency AMPAC-Net (G.G.), and ILLUQ 101133587 (M.L.). Any use of trade, firm, or product names is for descriptive purposes only and does not imply endorsement by the U.S. Government.

## Author contributions

K.M.W.A. conceived of the study, led the fieldwork, and wrote the paper. P.A. performed statistics. N.H. calculated chamber fluxes. C.E. was responsible for eddy-covariance data; E.E. assisted with the analysis. O.S., E.R., O.B., and K.M.W.A. conducted geochemical and microbial analyses on soil cores. B.M., N.P., and S.J. performed geophysical analyses. P.A., N.H., C.E., N.P., A.K., and M.S.B.-H. assisted with fieldwork. G.G. conducted remote sensing surveys of thermokarst mounds in the pan-Arctic. J.N. and M.L. performed thermokarst-mound talik modeling. All authors, including S.Z., commented on the analysis, interpretation, and presentation of the data, and were involved in the writing.

## Competing interests

The authors declare no competing interests.
