## [Peer Review File · Nature Communications]

Upland Yedoma taliks are an unpredicted source of atmospheric methaneREVIEWER COMMENTS

Reviewer #1 (Remarks to the Author):

The paper reports methane fluxes from ice-rich Yedoma deposits that experiences progressive thawing. Authors investigated seasonal and year-round CH₄ emissions from upland Yedoma taliks and observed increase of methane emissions during winter. They suggested that methanogenesis increased in the talik, and winter freezing of surface soils inhibites aerobic methane oxidation and enhances anoxia in the talik and boosts winter methanogenesis in taliks and CH₄ flux. Authors collected significant massive of data. The study is based on year-round chamber flux measurements from 2020 to 2023 at 25 upland sites characterized by thermokarst mounds and at 16 upland control sites lacking thermokarst mounds. Then, intensive measurements were conducted using eddy covariance, radiocarbon dating, geophysics, and borehole drilling. Authors used also modeling to predict CH₄ fluxes. The paper report soil analyses that includes determination of dry density; organic carbon and nitrogen concentrations; soil dissolved CH₄ and CO₂ concentrations; $\delta^{13}\text{C}$ values of CH₄ and CO₂; qPCR 737 analysis of the *mcrA* and *pmoA* genes, and 16S amplicon-based sequencing for characterization of the microbial community. The study describes significant amount of observational field data and will be interesting to other researchers who work in Arctic.

The authors are not very careful to details, almost all figures contain discrepancies with figure captions.

Fig 1. Explain why did you include sites specifically from Kotelny Island, Northeast Siberia and Tabaga, Central Yakutia if you do not provide field observation and measurements from those sites.

Figure 2. # c – showed year 2020, but in figure caption year indicated as 2015, please correct. # f – what reader should look for? How reader can distinguish between permafrost and talik? Where exactly top and bottom of the talik?

Fig. 5 Soil organic carbon – line is different in plot and figure caption

Figure 10. Indicate ice wedges by arrows.

Fig.S1 -What does show white circle on the world map? Show where Alaska is located. Remove “Alaska-scale”. What do you mean under “the pan-Arctic”?

Fig.S2 – a: what is difference between red Line A and blue Line D? b: Liquid water in figure and soil water in captions are those Volumetric Water Content? Change.

How BH1, BH6 and NMR boreholes are located with respect to each other? There is only 1 white circle for all 3 boreholes?

ohm-m – please spell out or explain

What does it mean “the depth to permafrost (6.8 m)”? Is permafrost located below 6.8 m? What depth does the permafrost start?

Fig. S3 – What layer the soil moisture was measured?

Fig. S6 – a: Move Legend out of graph. The plot and axes are shifted. Legend is confusing. What this: Hydro+Aceto? b: Not all colors in the graphs correspond to colors in legend. b-c: what cut-off percentage was for most abundant?

Fig. S7 – were measurements done at the same sites in 2021 and 2022? How other parameters in c correspond to a and b?

Fig. S8 There are large StDev for some samples. Please explain your conclusion that fluxes are not significantly different

Fig. S11 In figure captions, the a,b,c are not mentioned. Use them or remove from plots

Table S1 a – The site names are not distinguished good on Figure S1 (try different color). Where were control sites located?

Reviewer #2 (Remarks to the Author):

Remarks to the author:

First of all, I really enjoyed reading the manuscript! The authors present and analyse experimental data suggesting that dry Yedoma upland taliks are a significant, but previously unaccounted, methane source to the atmosphere. Furthermore, by developing a series of models considering different climate change scenarios, the authors predict that methane emissions could increase substantially in these ecosystems in the following decades. The data collection and analyses are substantial and sufficient to address the hypotheses. The combination of field CH₄ fluxes (i.e., chambers and EC), radio carbon dating, isotopic analyses, physicochemical analyses, hydrology monitoring, and modelling allow evaluating the major findings and conclusions. Major findings of this study include i) climate change driven thawing of dry Yedoma uplands, developing taliks, significantly increase methane emissions in the arctic; ii) most of these emissions occurred during winter due to a significant decrease in methanotrophic activity; and iii) widespread talik development in the arctic during the following decades will represent a major methane source to the atmosphere. The overall research, analyses, and conclusions are relevant and timely given the current climate crisis, as well as the urgent need to gain knowledge on the real contribution of Nature-based Solutions to achieve carbon neutrality. However, there are a couple of details about the methodology, statistical analysis, and data presentation that I would like to know if it would be possible for the authors to address/clarify. Once the comments are addressed, I consider that this manuscript

would be a relevant and necessary contribution to the field.

Major comments

1. L92-93. The authors indicate that “Methanogenesis in thawing Yedoma permafrost is active year-round” and it is suggested that methanotrophy is almost completely abated during winter. This, in conjunction with in situ methane emission measurements, removing top soil layers (Fig. 6), support the seasonality of the methane biofilter hypothesis. I agree completely with the author about the importance of methanotrophy. Actually, in the data, methanotrophy seems to be occurring during winter as well, as show in Fig. 5b (pmoA gene expression). This suggests that, as in other ecosystems, and as recognized by the authors, the overall methane emissions observed are the result of a balance between methanogenesis and methanotrophy occurring simultaneously. As consequence, positive (methanogenesis dominated) and close to zero-negative (methanotrophy dominated) emissions are equally relevant to evaluate the methanogenesis-methanotrophy relationship and the potential impact of climate change in the ecosystem’s carbon balance. Unfortunately, throughout the document (Fig.3; Fig.4, FigS4) close to zero and negative emissions were discarded from figures, presenting results as if all fluxes were positive. Can the author please:

1. Explain which was the rationale behind discarding these data from the figures?, this is particularly important as this data seems to represent more than one third of the data on each site. This bias perception towards an only net positive flux scenario.
2. Clarify if the zero-negative fluxes were included as part of the statistical analyses?

In order to completely discard methanotrophic activity due to low temperature it would have been necessary to conduct respirometric assays

2. Please present statistics to support the statement in L95: “Extreme wintertime barometric pressure fluctuations pump methane out of taliks through frost-induced cracks”.

Unlike Forde et al (2019) where, by injecting CH₄ in situ, it is possible to observe a clear effect of barometric pumping in their Fig. 2, analysis of data presented by the authors in Figure 8 does not seem to suggest a clear influence of this process in the study site.

In the supplementary data (L411) is mentioned that: “Spikes in methane emission, particularly in winter, correlated with abrupt decreases in atmospheric pressure”; however, the authors do not present any correlation analyses to support this and the suggested correlation presented in Fig. 8 is not self-evident.

Minor comments

Abstract

L33. It may be important to specifying that it is “potential”

Main text

L63. Please check the references formatting. Some are in Nature style; some others remain as from a previous version (e.g. L77).

L67. Please be consistent with the units used to present fluxes. It is quite difficult to contextualize and contrast data if different units are used throughout the text, tables, and figures. For example, in L30 methane emissions are presented as $\text{g m}^{-2} \text{ yr}^{-1}$, in L67 as $\text{mg CH}_4 \text{ m}^{-2} \text{ d}^{-1}$, in Figure 3 as $\mu\text{mol CH}_4 \text{ m}^{-2} \text{ s}^{-1}$. Even among tables the authors use different units.

L77. According to Hugelius et al. (2014) (Table 6), the Permafrost SOC stocks represent 181 Pg for Yedoma region, equivalent to 14 % of the Total permafrost region. The 407 Pg figure, and the associated 24 %, is a citation from Tarnocai et al (2009). I suggest using the most up to date estimates if the authors have already cited Hugelius, please be consistent with sources and citations.

L94-L96. These two sentences seem disjointed.

Figure. 1g. Please check that all figures are fully visible.

Figure. 2c,f. Please check that all figures are fully visible. In addition, in panel c, please clarify what do Line A and Line D represent?

L191-192. Can the authors provide statistics data to support these conclusions? Table S1b includes the data but does not include the statistics.

L194. Considering changing phrasing from indicating to “suggesting”. I consider that the experiment does not support such conclusion as only a single golf course was sampled. Furthermore, please check previous comment on the statistics to support the statement.

L197-198. Taking into account the large amount of supplementary data, please consider being more specific about which table or section includes the data and statistics supporting these conclusions. It is quite difficult to have to read throughout all the supplementary data to find the specific topic, table or figure the authors are referring to. Furthermore, if for this particular sentence the author makes reference to Table S1b, this table does not include statistical analyses but only descriptive statistics. Please provide statistical support for this: L197. “found no significant differences”.

Fig. 3 In panel a and b, what are the black dotted line models? In addition, please consider changing the colour of the arrows head to improve clarity, it is quite difficult to distinguish among all the lines.

L215-216. How are methane fluxes and soil moisture related? Based solely on Fig. 3, the relationship does not seem to be particularly strong. Are there more statistics to support this statement? Figure S3 does present some statistic data but the figure does not include the model or more appropriate parameters to evaluate the relationship between the two variables. Additionally, the data included as Supp Mat does not allow performing the analyses to evaluate the statement. From Figure 3 it seems as if there is not a strong correlation between the two variables. Despite the missing model, I appreciate seeing that the authors included the zero and negative fluxes, which are a substantial proportion of the data.

L230-233. Figure S4 does not support this statement. As the caption indicates, close to zero and negative fluxes were excluded. As previously mentioned, considering the large proportion of the data that was close to zero or negative, it cannot be excluded without presenting a clear rationale to support its removal. This is particularly important as methanotrophy and the role of dried soils through the stratigraphic profile are discussed throughout the manuscript as a major driver for large winter emissions. In addition, I suggest presenting the coefficient of determination instead of the correlation coefficient.

L233-239. These conclusions are based upon the analysis presented in Fig S4. However, correct me if I'm wrong, but such analysis does not include negative and or close to zero data, which is an integral part of the dataset.

L246-247. Are there statistics to support this statement? Figure S5 presents the data but does not present sufficient information to evaluate the significance of the models or the correlation between thermokarst-mound topography and water saturation.

L274. This figure only presents positive fluxes. Even though it is mentioned in the caption, the figure does not capture the actual variability in the fluxes.

L282-283. The paradigm regarding the typical methanogenesis-methanotrophy activity through the stratigraphic profile does not apply in many cases. For instance, in tropical peatlands it is the opposite, as labile substrates are available at the top layers while more recalcitrant substrates difficult methanogenesis at the deeper layers (10.1016/j.soilbio.2016.08.017).

L362. Is it possible that the flux increase is related to soil disturbance (e.g. collapse of pores and creation of fractures in soil matrix) rather than exclusively removing methanotrophic strata? Figure 6 does not have error bars, how many fluxes per site/depth did you take per date? Are there statistics to support this increase?

L367-370. Was there a significant difference between the treatments (i.e., thaw, summer, winter)? Table S4 does not include statistical analyses.

Fig.6 Revise Fig 6 panel b. There are data points out of the graph.

Fig. 7 Considering the temperatures reported in Table S4a. During winter, temperatures at the surface and at 10 cm are below 0° Celsius. At this temperature methanogenesis activity should also decrease substantially to the point of being almost negligible. Would it be possible that physical processes drive the larger emissions during winter and that the methane was produced during the summer and autumn? To test this hypothesis, it would have been necessary to run ex situ incubations at low temperatures for samples from different sections of the stratigraphic profile. In previous studies, it has been observed that both methanogenesis and methanotrophy are significantly impacted by low temperatures. (10.1016/j.scitotenv.2018.04.283).

L412. Please use of units consistently. These are not the units used in Figure 8.

L413. The mean \pm SD seems out of place. Either way, why is it that the authors decided to use SD in this section but s.e.m. in the Supp Matt table S4a? On L432 the authors use mean \pm standard error.

L439-440. Is there literature supporting this phenomenon?

L475. Please change to km² for consistency.

L802-803. Photographs and video are insufficient to replicate of the experiment. The video does not show the process.

April 23, 2024

Letter to the Referees

The reviewers' comments on our manuscript (NCOMMS-23-64233), previously entitled, "*Dry Yedoma upland taliks are an unpredicted source of atmospheric methane*," were helpful. We have responded to these comments in the revised manuscript (reviewer comments repeated here in bold) and attempted to address their concerns in the following ways:

Please note, in response to USGS reviewer comment 2 concerning "dry" being too qualitative a term, we have revised the title to, "*Upland Yedoma taliks are an unpredicted source of atmospheric methane.*"

REVIEWER COMMENTS

Reviewer #1 (Remarks to the Author):

The paper reports methane fluxes from ice-rich Yedoma deposits that experiences progressive thawing. Authors investigated seasonal and year-round CH₄ emissions from upland Yedoma taliks and observed increase of methane emissions during winter. They suggested that methanogenesis increased in the talik, and winter freezing of surface soils inhibites aerobic methane oxidation and enhances anoxia in the talik and boosts winter methanogenesis in taliks and CH₄ flux. Authors collected significant massive of data. The study is based on year-round chamber flux measurements from 2020 to 2023 at 25 upland sites characterized by thermokarst mounds and at 16 upland control sites lacking thermokarst mounds. Then, intensive measurements were conducted using eddy covariance, radiocarbon dating, geophysics, and borehole drilling. Authors used also modeling to predict CH₄ fluxes. The paper report soil analyses that includes determination of dry density; organic carbon and nitrogen concentrations; soil dissolved CH₄ and CO₂ concentrations; $\delta^{13}\text{C}$ values of CH₄ and CO₂; qPCR 737 analysis of the *mcrA* and *pmoA* genes, and 16S amplicon-based sequencing for characterization of the microbial community. The study describes significant amount of observational field data and will be interesting to other researchers who work in Arctic.

The authors are not very careful to details, almost all figures contain discrepancies with figure captions.

1) Fig 1. Explain why did you include sites specifically from Kotelny Island, Northeast Siberia and Tabaga, Central Yakutia if you do not provide field observation and measurements from those sites.

In the previous version of the manuscript we labeled Kotelny Island and Tabaga in map panel *a* because these two locations are mentioned in the legend as examples of remote-sensing detection of thermokarst mounds shown in panels *b* and *c*. Our goal was to inform readers of the locations

of these remote-sensing sites. To avoid confusion, since no field work was conducted at these sites, we removed the site-name labels from the figure and replaced the location markers with asterisks.

2) Figure 2. # c – showed year 2020, but in figure caption year indicated as 2015, please correct. # f – what reader should look for? How reader can distinguish between permafrost and talik? Where exactly top and bottom of the talik?

We thank the reviewer for pointing out the typo in our figure legend. In an earlier iteration of this figure we had used a 2015 image. Later we updated the image to reflect a point in time closer to our study period (year 2020), but we neglected to update the legend. We have now corrected this oversight to ensure that the figure label and legend both consistently show year 2020.

We also relabeled panel f so that the arrows are unidirectional pointing to intersection of the base of talik and top of permafrost. We improved the syntax of the legend to more clearly indicate that the ground surface is shown in panel e, while talik base and permafrost top are shown in f. The purpose of including these photos was to demonstrate the dry soil conditions. To improve clarity, we added the soil moisture values to the revised legend. See also USGS Reviewer comment 5.

3) Fig. 5 Soil organic carbon – line is different in plot and figure caption

We changed the brown dashed line to a solid brown line so that soil organic carbon is consistent in the plot and legend.

4) Figure 10. Indicate ice wedges by arrows.

We added arrows to indicate ice wedges.

5) Fig.S1 -What does show white circle on the world map? Show where Alaska is located. Remove “Alaska-scale”. What do you mean under “the pan-Arctic”?

To avoid redundancy and confusion and because the pan-arctic extent of Yedoma soils is already shown in Figure 1, we removed the world map from Fig. S1.

6) Fig.S2 – a: what is difference between red Line A and blue Line D?

The red and blue lines are both geophysical ERT observation locations, but with different orientations. We ensured that the lines were clearly labeled and identified as such in both Fig. S2 and in the main manuscript Figure 2.

b: Liquid water in figure and soil water in captions are those Volumetric Water Content? Change.

Yes, panel b is volumetric water content. We have revised the x-axis title to reflect this.

How BH1, BH6 and NMR boreholes are located with respect to each other? There is only 1 white circle for all 3 boreholes?

At the scale of map panel *a*, a single dot represents the three boreholes, which are all within ~1-m of each other. We revised the figure legend to explain this.

ohm-m – please spell out or explain

We spelled out “ohm-meter” as the units for panel *e*.

What does it mean “the depth to permafrost (6.8 m)”? Is permafrost located below 6.8 m? What depth does the permafrost start?

We clarified in the legend that the top of permafrost (start of permafrost) is at 6.8 m.

7) Fig. S3 – What layer the soil moisture was measured?

The previous Fig. S3 has been removed from the manuscript and replaced with a revised version of Fig. 3 (see Reviewer 2, main comment 1a). All data previously shown in Fig. S3 are now shown in Fig. 3. We revised the Fig. 3 legend to explain that soil moisture was measured at 12 cm.

8) Fig. S6 – a: Move Legend out of graph. The plot and axes are shifted. Legend is confusing. What this: Hydro+Aceto? b: Not all colors in the graphs correspond to colors in legend. b-c: what cut-off percentage was for most abundant?

We thank the reviewer for these suggestions. We revised the figure in the following ways: a) We moved the legend out of the graph and replotted the data so that the axes align exactly. We also removed “Hydro+Aceto” from the legend and added a phrase to the caption to explain that *Methanosarcina* archaea includes both hydrogenotrophic and acetoclastic pathways.

b) We checked that the colors in the graphs correctly correspond to the colors in the legend. The darkest blue, representing *Crenarchaeota Nitrosopumilales*, does not appear in core BH6 graph because this phylum was absent in the BH6 core.

b-c) We revised the figure caption to state that the threshold used for ASVs relative abundance in panels b and c was >1%.

9) Fig. S7 – were measurements done at the same sites in 2021 and 2022? How other parameters in c correspond to a and b?

We revised the figure caption to clarify that the measurements in *a*, *b* and *c* were all done at the same site, which was the eddy covariance tower site at North Star Yedoma. Parameters in *c* were measured in the soil, while parameters in *a* and *b* were measured in the atmosphere, but all were at the same site.

10) Fig. S8 There are large StDev for some samples. Please explain your conclusion that fluxes are not significantly different

To be consistent with the way chamber and tower data have been compared in the literature, we revised the manuscript to present the data in a 1:1 scatterplot format (please see revised Fig. S7, panel b) (Kade et al. 2012, doi:10.1029/2012JG002065; Podgrajsek et al. 2014,

doi.org/10.5194/bg-11-4225-2014; Morin et al. 2017, doi.org/10.1016/j.agrformet.2017.01.022). Linear regression on the raw data did not meet the assumption of homoscedasticity, so we log-transformed the data: Linear regression through the origin of chamber and tower fluxes ($r^2=.89$, $F(1,14)=120$, $p<.001$). We decided not to present the results of linear model because of difficulties in inference from regression on log-transformed data. However, from the distribution of the points along the 1:1 line, it is evident that there is no systematic bias based on chamber vs tower methods. We revised the main text and Fig. S7 caption to reflect this conclusion. We also changed standard deviations to standard errors to be consistent with the format of uncertainty representation throughout the rest of the manuscript (see Reviewer 2, comment 25).

11) Fig. S11 In figure captions, the a,b,c are not mentioned. Use them or remove from plots.

We added the panel labels to the caption text (revised Fig. S12).

12) Table S1 a – The site names are not distinguished good on Figure S1 (try different color). Where were control sites located?

We revised Fig. S1 to improve legibility of the site IDs. We did not plot the controls because they are adjacent to the thermokarst-mounds sites and would be undiscernible on the map at this scale; however, we added a sentence to the caption stating that the coordinates for the adjacent control sites are provided in Table S1a.

Reviewer #2 (Remarks to the Author):

Remarks to the author:

First of all, I really enjoyed reading the manuscript! The authors present and analyse experimental data suggesting that dry Yedoma upland taliks are a significant, but previously unaccounted, methane source to the atmosphere. Furthermore, by developing a series of models considering different climate change scenarios, the authors predict that methane emissions could increase substantially in these ecosystems in the following decades. The data collection and analyses are substantial and sufficient to address the hypotheses. The combination of field CH₄ fluxes (i.e., chambers and EC), radio carbon dating, isotopic analyses, physicochemical analyses, hydrology monitoring, and modelling allow evaluating the major findings and conclusions. Major findings of this study include i) climate change driven thawing of dry Yedoma uplands, developing taliks, significantly increase methane emissions in the arctic; ii) most of these emissions occurred during winter due to a significant decrease in methanotrophic activity; and iii) widespread talik development in the arctic during the following decades will represent a major methane source to the atmosphere. The overall research, analyses, and conclusions are relevant and timely given the current climate crisis, as well as the urgent need to gain knowledge on the real contribution of Nature-based Solutions to achieve carbon neutrality. However, there are a couple of details about the methodology, statistical analysis, and data presentation that I would like to know if it would be possible for the authors to address/clarify. Once the

comments are addressed, I consider that this manuscript would be a relevant and necessary contribution to the field.

Major comments

1. L92-93. The authors indicate that “Methanogenesis in thawing Yedoma permafrost is active year-round” and it is suggested that methanotrophy is almost completely abated during winter. This, in conjunction with in situ methane emission measurements, removing top soil layers (Fig. 6), support the seasonality of the methane biofilter hypothesis. I agree completely with the author about the importance of methanotrophy. Actually, in the data, methanotrophy seems to be occurring during winter as well, as show in Fig. 5b (pmoA gene expression). This suggests that, as in other ecosystems, and as recognized by the authors, the overall methane emissions observed are the result of a balance between methanogenesis and methanotrophy occurring simultaneously. As consequence, positive (methanogenesis dominated) and close to zero-negative (methanotrophy dominated) emissions are equally relevant to evaluate the methanogenesis-methanotrophy relationship and the potential impact of climate change in the ecosystem’s carbon balance. Unfortunately, throughout the document (Fig.3; Fig.4, FigS4) close to zero and negative emissions were discarded from figures, presenting results as if all fluxes were positive. Can the author please:

1a) Explain which was the rationale behind discarding these data from the figures?, this is particularly important as this data seems to represent more than one third of the data on each site. This bias perception towards an only net positive flux scenario.

We thank the reviewer for the request to include the zero and negative emissions in our figures. Due to the awkwardness of plotting zero and negative values on a log scale (Fig. 3a-d, Fig. 4) and due to relatively low values of the negative fluxes compared to positive fluxes (Fig. 3e) we had opted to describe these data in the figure captions and show them in previous manuscript Fig. S3. However, we appreciate the reviewer’s point that providing a visualization of the zero and negative fluxes alongside the positive fluxes will give a more comprehensive presentation of the distribution of measured fluxes to readers who may not pay as much attention to the Supplementary Information. We removed Figure S3 and have revised Figures 3 and 4 to show all of the fluxes. In the caption we explain that in order for the relatively low negative fluxes to be distinguishable in Fig. 3f we had to multiply them by a factor 10.

1b) Clarify if the zero-negative fluxes were included as part of the statistical analyses?

Yes, the zero and negative fluxes were included as part of the statistical analyses in the previous version of the manuscript. They are also included in the analysis of the revised manuscript. The only exception is the regression analysis in Fig. S3 (previous Fig. S4), which we explain below (see comment 15).

1c) In order to completely discard methanotrophic activity due to low temperature it would have been necessary to conduct respirometric assays.

We do not claim that methanotrophy is completely absent in winter; rather it appears to be lower than summer. We revised the manuscript to provide support for this statement in the context of the stable isotope data and pmoA expression, which was 30-fold lower in the winter BH6 core ($1.3 \times 10^6 \pm 3.27 \times 10^5$ copies/1gr soil, at 100 cm depth) compared to the summer BH1 core ($4.05 \times 10^7 \pm 1.02 \times 10^6$; a decrease of 30-fold at 68 cm depth) (Fig. 5b, Supplementary Discussion 5.4). This wintertime limitation of methanotrophy was likely due to the reduced capacity of oxygen to penetrate ice-filled soil pores in the seasonal frost layer and metabolic response of methanotrophs to cold temperature (Dunfield et al. 1993, doi.org/10.1016/0038-0717(93)90130-4; Sepulveda-Jauregui et al. 2018, doi.org/10.1016/j.scitotenv.2018.04.283; Zona et al. 2016, doi/10.1073/pnas.1516017113; Howard et al. 2020, doi.org/10.5194/bg-17-4025-2020).

We also added a sentence to Supplementary Discussion 5.2 stating, “*soil respirometric assays would also be useful to determine temperature, redox and nutrient controls on methanogenesis and methanotrophy in upland taliks.*”

2) Please present statistics to support the statement in L95: “Extreme wintertime barometric pressure fluctuations pump methane out of taliks through frost-induced cracks”. Unlike Forde et al (2019) where, by injecting CH₄ in situ, it is possible to observe a clear effect of barometric pumping in their Fig. 2, analysis of data presented by the authors in Figure 8 does not seem to suggest a clear influence of this process in the study site. In the supplementary data (L411) is mentioned that: “Spikes in methane emission, particularly in winter, correlated with abrupt decreases in atmospheric pressure”; however, the authors do not present any correlation analyses to support this and the suggested correlation presented in Fig. 8 is not self-evident.

We thank the reviewer for suggesting a more in-depth analysis of the relationship between barometric pressure and ebullition. In the revised manuscript we conducted statistical analyses to support our statements, added Supplementary Figures S8 and S9, and provided a more comprehensive discussion of the relationship (Supplementary Discussion 5.2).

Minor comments

Abstract

3) L33. It may be important to specifying that it is “potential”

We added “potential” to this sentence.

Main text

4) L63. Please check the references formatting. Some are in Nature style; some others remain as from a previous version (e.g. L77).

We checked that all references are called out in the superscript style of *Nature Communications*.

5) L67. Please be consistent with the units used to present fluxes. It is quite difficult to

contextualize and contrast data if different units are used throughout the text, tables, and figures. For example, in L30 methane emissions are presented as g m⁻² yr⁻¹, in L67 as mg CH₄ m⁻² d⁻¹, in Figure 3 as μmol CH₄ m⁻² s⁻¹. Even among tables the authors use different units.

We appreciate this comment and have revised the manuscript, including all tables and figures, to present our results consistently in units of mg m⁻² d⁻¹. Two exceptions are the comparison of chamber fluxes to tower fluxes (Fig. S7) and the comparison of annual emissions from NSY to those of other upland and wetland systems presented in the literature (Fig. 9). In the case of Fig. S7, we followed the conventional nomenclature in the field, which is to compare chamber and tower methods using the original units of measurement (μmol CH₄ m⁻² s⁻¹) rather than upscaling to daily rates. In the case of the circumpolar ecosystem emissions comparison (Fig. 9), we explained that we've summarized our data on an annual basis to compare to emissions reported for other ecosystems, the synthesis of which is only available on an annual, not daily basis (Treat et al. 2018, doi.10.1111/gcb.14137).

6) L77. According to Hugelius et al. (2014) (Table 6), the Permafrost SOC stocks represent 181 Pg for Yedoma region, equivalent to 14 % of the Total permafrost region. The 407 Pg figure, and the associated 24 %, is a citation from Tarnocai et al (2009). I suggest using the most up to date estimates if the authors have already cited Hugelius, please be consistent with sources and citations.

We have taken a careful look at the literature pertaining to permafrost soil organic carbon inventories and revised the manuscript to cite the most up-to-date estimate of the Yedoma domain permafrost soil organic carbon (SOC) pool (327-466 Pg C), which is from Strauss et al. (2017, doi.org/10.1016/j.earscirev.2017.07.007). We followed the example of Strauss et al. to place Yedoma SOC in context of the circumpolar permafrost soil organic carbon pool from Hugelius et al. (2014) (1307 Pg), and still found that yedoma contributes a disproportionately large fraction (>25%). Strauss et al. (2017) reached the same conclusion based on total soil organic carbon pools (i.e., the numbers we present) as well as on just the perennially frozen fractions of these pools. We have revised the manuscript to ensure we clearly cite the most updated carbon pool estimates and their sources.

7) L94-L96. These two sentences seem disjointed.

We joined the two sentences.

8) Figure. 1g. Please check that all figures are fully visible.

We revised Figure 1 to improve legibility, including a change from white to black font in Figure 1g.

9) Figure. 2c,f. Please check that all figures are fully visible. In addition, in panel c, please clarify what do Line A and Line D represent?

We thank the reviewer for suggestions to improve this figure. We have ensured that the labels in the revised figure are fully visible and we specified in the revised legend that Lines A and D are the locations of the geophysical observations.

10) L191-192. Can the authors provide statistics data to support these conclusions? Table S1b includes the data but does not include the statistics.

In the revised manuscript we performed statistical analysis on the data presented in Table S1b. First, we performed a two-way ANOVA on rank-transformed data in Table S1b to analyze the effect of study site and type (thermokarst mound vs control) on plot-scale chamber-based methane fluxes. The two-way ANOVA revealed no significant interaction between study sites and type (thermokarst-mound vs. control) on plot-scale chamber methane fluxes ($F(10,609)=1.32$, $p=.21$), but simple main effects showed that emissions from thermokarst mounds were higher than from the adjacent controls ($p<.0001$). Then, among the 26 thermokarst-mound study sites, two-way ANOVA on rank-transformed data revealed no interaction between site and season on methane flux ($F(3,1112)=2.0$, $p=.11$), but simple main effect of site was significant ($p<.001$). We revised Table S1b to present site-specific methane emissions in mean rank order from highest to lowest for thermokarst-mounds. We also added Figure 3e to show methane fluxes (mean, standard error) for the 26 sites including all positive, zero and negative values. We report that fluxes at NSY fell within the range of 23 other methane-emitting Alaskan upland thermokarst-mound sites (Fig. 3a,b,e; Table S1b) [NSY: 110 ± 24 mg CH₄ m⁻² d⁻¹, mean \pm standard error (s.e.m.), $n=665$; Other extensive sites: 87 ± 38 mg CH₄ m⁻² d⁻¹, $n=450$].

11) L194. Considering changing phrasing from indicating to “suggesting”. I consider that the experiment does not support such conclusion as only a single golf course was sampled. Furthermore, please check previous comment on the statistics to support the statement.

During the revision process this phrase was removed from the manuscript. However, we carried out full statistical analysis to compare the NSY study site to other thermokarst mounds and presented results in the revised manuscript. See revised Figure 3, Table S1b, and comment 10.

12) L197-198. Taking into account the large amount of supplementary data, please consider being more specific about which table or section includes the data and statistics supporting these conclusions. It is quite difficult to have to read throughout all the supplementary data to find the specific topic, table or figure the authors are referring to. Furthermore, if for this particular sentence the author makes reference to Table S1b, this table does not include statistical analyses but only descriptive statistics. Please provide statistical support for this: L197. “found no significant differences”.

In general, we revised the manuscript to more carefully call out specific Supplementary Data locations referred to in the main text.

The specific main text sentence in question here refers to the analysis reported in Supplementary Results 4.1. In the revised manuscript we also performed statistical analysis to support our statement of “no significant differences.” A Kruskal-Wallis rank sum test indicated no significant difference in methane flux between unmanaged thermokarst-mounds and areas heavily impacted by soil and vegetation management, $X^2(1)=0.171$, $p=.679$.

13) Fig. 3 In panel a and b, what are the black dotted line models? In addition, please consider changing the colour of the arrows head to improve clarity, it is quite difficult to distinguish among all the lines.

In the previous version of this figure, the black dotted line models showed cumulative emissions. The lines have been removed from the revised figure, which now shows all the flux values (positives, negatives and zeros). In panel f, we followed the reviewer's suggestion to improve clarity by modifying the display of the arrows.

14) L215-216. How are methane fluxes and soil moisture related? Based solely on Fig. 3, the relationship does not seem to be particularly strong. Are there more statistics to support this statement? Figure S3 does present some statistic data but the figure does not include the model or more appropriate parameters to evaluate the relationship between the two variables. Additionally, the data included as Supp Mat does not allow performing the analyses to evaluate the statement. From Figure 3 it sees as if there is not a strong correlation between the two variables. Despite the missing model, I appreciate seeing that the authors included the zero and negative fluxes, which are a substantial proportion of the data.

We revised Figure 3 to show surface soil moisture compared to all fluxes (positive, zero and negative) for the extensive thermokarst mound sites and NSY. These data were previously shown in Supplementary Figure S3, which has been removed from the revised manuscript to avoid redundancy. In the revised manuscript we also provide a more comprehensive analysis of the statistical relationships between surface soil moisture and methane flux. The reviewer is correct that the relationship is not terribly strong (owing to subsurface factors described later in the manuscript), but nonetheless, it was significant: *extensive thermokarst mounds*: Pearson correlation coefficient $r(273)=.33$, $p<.001$; *NSY*: $r(314)=.51$, $p<.001$). We added a new section (Supplementary Results 4.2) to present moisture thresholds for the zero and negative fluxes and discuss the implications of wider surface soil moisture conditions at the extensive thermokarst mounds despite similar methane fluxes as NSY.

15) L230-233. Figure S4 does not support this statemen. As the caption indicates, close to zero and negative fluxes were excluded. As previously mentioned, considering the large proportion of the data that was close to zero or negative, it cannot be excluded without presenting a clear rationale to support its removal. This is particularly important as methanotrophy and the role of dried soils through the stratigraphic profile are discussed throughout the manuscript as a major driver for large winter emissions. In addition, I suggest presenting the coefficient of determenation instead of the correlation coefficient.

Per the reviewer's suggestion, we revised the manuscript to present all positive, zero and negative flux data. These data were included in our statistical analyses and they appeared previously in Fig. S3 and all tables. In the revised manuscript, we now consistently show all the values in the main manuscript display items and Supplementary Information. The only exception is the regression analysis presented in revised Fig. S3 (previous manuscript Fig. S4). It is difficult to capture these zero and negative fluxes on a log scale with the positives, and we were

limited since our relationships were strongest to the log-scaled positive fluxes. We did explore linear relationships to a variety of our geophysical water content metrics (shallow and deep averages, maxes, etc.) while keeping the zeros and negatives included. Unfortunately, these relationships are not as strong and can be explained. First, the negative and zero fluxes do tend to occur at lower water contents, but they still span a range up to 50% VWC, as we've shown in the surface soil moisture analysis as well (Fig. 3). So, there may be a critical threshold below which negative/zero flux is observed (see new Supplementary Results section 4.2), but that does not follow the continuous log-scaled relationship along with positive fluxes. Second, methane uptake may be influenced by very shallow water content at a spatial scale below what we can resolve clearly with our geophysical methods. Third, shallow dry conditions can correlate both with methane uptake and with enhanced positive flux in our proposed scenario with deeper elevated water content (due to easier escape pathways for methane generated at depth). The latter is what we believe we are best capturing with our geophysical measurements—we think we are seeing the role of deeper saturated conditions driving positive fluxes, but net flux is potentially mitigated by shallow conditions. Lastly, the sample size we are working with is likely too small for us to meaningfully account for negative and zero fluxes. Out of 36 flux locations that fall within 2 meters of our ERT transects, only 4 are zero and 2 are negative. Given these points, we feel justified in our current approach of focusing on the positive fluxes. In addition to edits in the Supplementary Results, we revised the main text to support this decision, stating “*due to the low number of zero and negative fluxes recorded within 2 m of the ERT transects— $n=4$ and $n=2$, respectively out of 36 total data points—relationships were dominantly driven by the positive methane fluxes*”. We also added the coefficient of determination to Fig. S3 (previous manuscript Fig. S4).

16) L233-239. These conclusions are based upon the analysis presented in Fig S4. However, correct me if I'm wrong, but such analysis does not include negative and or close to zero data, which is an integral part of the dataset.

In the previous comment (comment 15), we provided a justification for the exclusion of six data points from the Fig. S3 (previous manuscript Fig. S4). Please note, however, that the six points in question were included in the main manuscript (Figures 3 and 4) and in all the rest of the statistical analyses in our manuscript. We also revised the main text to explain that the conclusions presented in this section based on Fig. S3 are consistent with surface-soil moisture and methane relationships to microtopography, an analysis based on much larger sample sizes and statistical analyses including all data (positive, zero and negative fluxes) (Table S3).

17) L246-247. Are there statistics to support this statement? Figure S5 presents the data but does not present sufficient information to evaluate the significance of the models or the correlation between thermokarst-mound topography and water saturation.

We revised Figure S4 (previous manuscript Fig. S5) to address the USGS reviewer's request to simplify this figure (USGS comment 21), while also satisfying this reviewer's request to provide statistics to support the statement. We added an F-test to identify which spectral peaks are statistically significant. This didn't change the overall findings or interpretations. We still see signals in the water content consistent with thermokarst mound topography, and these signals are

significant with 90% confidence. We revised the SI text to reflect this change and explain the new method.

18) L274. This figure only presents positive fluxes. Even though it is mentioned in the caption, the figure does not capture the actual variability in the fluxes.

We agree with the reviewer that a comprehensive presentation of all fluxes is important. We revised this figure to show all the fluxes, positive, zero and negative. The result is that negative and zero fluxes comprise a small fraction of the observations and are small in magnitude compared to the positive fluxes.

19) L282-283. The paradigm regarding the typical methanogenesis-methanotrophy activity through the stratigraphic profile does not apply in many cases. For instance, in tropical peatlands it is the opposite, as labile substrates are available at the top layers while more recalcitrant substrates difficult methanogenesis at the deeper layers (10.1016/j.soilbio.2016.08.017).

We thank the reviewer for pointing out the need to qualify this statement since tropical peatlands provide an interesting and opposite dynamic based on substrate availability alone. As an aside, substrate is also a driver of methanogenesis-depth dynamics in thermokarst environments, with young (^{14}C -enriched), labile substrates available in surface sediments and ^{14}C -depleted substrates supplied through permafrost thaw at depth (Heslop et al. 2015, doi:10.5194/bg-12-4317-2015). Since the current manuscript focuses on soil moisture and water table controls on methane dynamics, we revised the text to qualify our statement, placing the paradigm in the context of northern wetlands, “*Although NSY is not a wetland, both summer and winter soil geochemical and microbial profile data agreed with the current paradigm for northern wetlands that in situ methanogenesis occurs in deeper, anaerobic soils, and dry surface soils facilitate aerobic methanotrophy.*”

20) L362. Is it possible that the flux increase is related to soil disturbance (e.g. collapse of pores and creation of fractures in soil matrix) rather than exclusively removing methanotrophic strata? Figure 6 does not have error bars, how many fluxes per site/depth did you took per date? Are there statistics to support this increase?

We agree that digging soil pits and drilling boreholes certainly disturbed the soil structure and could have contributed to the elevated fluxes. We revised the manuscript to acknowledge this. We also revised Figure 6 to include error bars and we stated in the caption that the error bars represent the standard error of two or three measurements per depth. Finally, we added the control site data to this figure and we performed comprehensive statistical analyses to support our conclusions that thermokarst-mound boreholes and soil pits had higher emissions than control-site boreholes and soil pits, and that among thermokarst-mound profiles, methane emissions increased with depth. In contrast, the control-site boreholes and soil pits had negative fluxes, which tended more negative with depth.

In Supplementary Discussion 5.1 we added the following text: “*Collapse of soil pores and possible creation of new fractures in the matrix by our disturbances may have contributed to some of the elevated fluxes, but the disturbance itself did not introduce any directional bias in*

fluxes. This is evident from seven control site observations where we did not observe an increase in methane flux with depth following disturbance to surface soils. Also, among the thermokarst-mound soil pits, occasionally we observed a decrease (not increase) in flux. Altogether, the observations imply that soil methane concentrations increase with depth in thermokarst-mound taliks and that surface soils act as a strong biofilter.”

21) L367-370. Was there a significant difference between the treatments (i.e., thaw, summer, winter)? Table S4 does not include statistical analyses.

We performed a Kruskal-Wallis rank sum test followed by the Wilcoxon pairwise comparison test to show statistically significant differences in eddy-covariance measured methane and carbon dioxide by season within different study years (Table S4a). We added the statistical results to revised Table S4a.

22) Fig.6 Revise Fig 6 panel b. There are data points out of the graph.

We revised Figure 6 to more clearly show all the flux values using a log scale without having any points outside the graph.

23) Fig. 7 Considering the temperatures reported in Table S4a. During winter, temperatures at the surface and at 10 cm are below 0° Celsius. At this temperature methanogenesis activity should also decrease substantially to the point of being almost negligible. Would it be possible that physical processes drive the larger emissions during winter and that the methane was produced during the summer and autumn? To test this hypothesis, it would have been necessary to run ex situ incubations at low temperatures for samples from different sections of the stratigraphic profile. In previous studies, it has been observed that both methanogenesis and methanotrophy are significantly impacted by low temperatures. (10.1016/j.scitotenv.2018.04.283).

We agree that methanogenesis and methanotrophy are both impacted by low temperatures. As the reviewer pointed out, incubation studies demonstrate that between these two competing processes, methanogenesis is the more temperature sensitive (Dunfield et al. 1993, doi.org/10.1016/0038-0717(93)90130-4, Sepulveda-Jauregui et al. 2018, doi.10.1016/j.scitotenv.2018.04.283). If our study environment was limited to a single soil depth zone where methanogenesis and methanotrophy were in competition with each other under the same temperature regime simultaneously, then methanotrophy should outcompete methanogenesis in winter. However, NSY is not like most arctic soil profiles studied so far in that the talik is many meters deeper leading to important seasonally different thermal regimes throughout the profile. In Figure 5, we observed a peak in mcrA gene expression in summer soil profile at 150 cm (a time when surface soils were warmest). This peak migrated down to ~300 cm in winter, a time when surface soils are coldest, but thermal lag warmed the deeper sediments. Previous work showing that methane is produced (and oxidized anaerobically) in deep Yedoma taliks beneath thermokarst lakes at temperatures close to 0° C and that a thermal lag in summer heat propagation extends into the talik warming it at depth in winter provides evidence in support of our conclusion that methanogenesis continues at depth in the NSY talik

throughout winter (Heslop et al. 2015, doi.org/10.1016/j.earscirev.2020.103365; Winkel et al. 2019, doi.org/10.1088/2515-7620/ab1042). Since the summer soil profile data were limited to 3-m and we cannot compare the seasonal dynamics down to 7-m, we revised the manuscript to tone down the language about enhanced methanogenesis in winter. Instead we acknowledge that methanogenesis continues through winter (instead of being enhanced). We agree with the reviewer that respirometric assays would be useful to assess such a statement and have recommended in the revised manuscript assays be used in future research (Supplementary Information 5.2).

A leading cause of higher methane emissions in winter is, however, reduced methanotrophy (and as illustrated in Figures 7 and 8). This is evident by pmoA expression, which is ~30-fold lower in winter compared to summer (Figure 5). We revised the text to clarify these points. We added several literature references in support of temperature sensitivity, and we added a small enlargement square to Figure 5b to more clearly demonstrate that there is a small, but decreased peak in methanotrophy (pmoA expression) in winter compared to summer. In Supplementary Discussion 5.2 we also recommended future work focused on respirometric assays related to temperature sensitivities of methanogenesis and methanotrophy.

The reviewer also raised a question about the timing of summer-produced methane emission to the atmosphere. We also cannot exclude the possibility that some of the methane emitted in winter was produced in summer, and that its emission is enhanced in winter by physical processes, such as barometric pressure pumping (see comment 2). We revised the main text regarding wintertime pressure pumping to express this possibility.

24) L412. Please use of units consistently. These are not the units used in Figure 8.

We moved the reference to Figure 8 to a more appropriate location and revised the manuscript to ensure consistent use of units. Please see comment 5 for the explanation of two exceptions.

25) L413. The mean \pm SD seems out of place. Either way, why is it that the authors decided to use SD in this section but s.e.m. in the Supp Matt table S4a? On L432 the authors use mean \pm standard error.

We revised the manuscript to consistently use standard error of the mean, defining also the sample size n.

26) L439-440. Is there literature supporting this phenomenon?

We revised the manuscript to explain that to our knowledge this mechanism has not been explored in permafrost soils, but air circulation into porous rock beds is more vigorous in interior Alaska in winter (Goering 2003, doi.org/10.1061/(ASCE)0887-381X(2003)17:3(119) and thermal convection is known to enhance gas fluxes within fractures in other dryland soils (DeCarlo and Caylor 2003, https://doi.org/10.1016/j.geoderma.2020.114478). See also USGS Reviewer comment 18.

27) L475. Please change to km² for consistency.

We replaced “square kilometers” with “km²”.

28) L802-803. Photographs and video are insufficient to replicate of the experiment. The video does not show the process.

We removed the phrase “show the process” from this sentence and instead clarified that the reference provides a photo of the sampling and video of the turf bubbles. The sampling process itself is described in detail in the revised manuscript methods section.

We thank the reviewers for their helpful remarks, which have caused us to improve the paper. We hope that the revised manuscript will now be found suitable for publication in Nature Communications.

REVIEWERS' COMMENTS

Reviewer #1 (Remarks to the Author):

The manuscript was revised and authors responded on all my comments. The only comment is about panel f in Figure 2. It looks like picture sifted (as it displayed on my computer) and it does not show correct image.

Reviewer #2 (Remarks to the Author):

I would like to thank the authors for their effort and time in addressing all the observations included in the revision of the manuscript. This new version of the manuscript presents a clearer narrative with well-supported conclusions. The revised results now provide sufficient information to evaluate the scope and limitations of the conclusions, and the methodology is thoroughly explained. I have no additional observations. The manuscript is now suitable to be published in Nature Communications.

June 20, 2024

Letter to the Referees

We thank the reviewers for their feedback on our manuscript (NCOMMS-23-64233A) entitled, "*Upland Yedoma taliks are an unpredicted source of atmospheric methane.*" We have responded to their comments in the revised manuscript (reviewer comments repeated here in bold) and attempted to address their concerns in the following ways:

REVIEWER COMMENTS

Reviewer #1 (Remarks to the Author):

The manuscript was revised and authors responded on all my comments. The only comment is about panel f in Figure 2. It looks like picture sifted (as it displayed on my computer) and it does not show correct image.

We checked the source imagery. The correct fragment is displayed. We have provided Figure 2 as a bitmap with this submission with the expectation that the image will not be shifted or incorrectly displayed.

Reviewer #2 (Remarks to the Author):

I would like to thank the authors for their effort and time in addressing all the observations included in the revision of the manuscript. This new version of the manuscript presents a clearer narrative with well-supported conclusions. The revised results now provide sufficient information to evaluate the scope and limitations of the conclusions, and the methodology is thoroughly explained. I have no additional observations. The manuscript is now suitable to be published in Nature Communications.

We thank the reviewers for their thorough consideration through this process. Their helpful remarks caused us to improve the paper and we are grateful that it is now found suitable for publication in Nature Communications.